# Cooperative Stochastic Bandits with Asynchronous Agents and Constrained Feedback

**Lin Yang, Yu-Zhen Janice Chen**
University of Massachusetts Amherst
{linyang,yuzhenchen}@cs.umass.edu

**Stephen Pasteris**
University College London
stephen.pasteris@gmail.com

**Mohammad H. Hajiesmaili**
University of Massachusetts Amherst
hajiesmaili@cs.umass.edu

**John C. S. Lui**
Chinese University of Hong Kong
cslui@cse.cuhk.edu.hk

**Don Towsley**
University of Massachusetts Amherst
towsley@cs.umass.edu

## Abstract

Motivated by the scenario of large-scale learning in distributed systems, this paper studies a scenario where $M$ agents cooperate together to solve the same instance of a $K$-armed stochastic bandit problem. The agents have limited access to a local subset of arms and are asynchronous with different gaps between decision-making rounds. The goal is to find the global optimal arm, and agents are able to pull any arm; however, they can only observe the reward when the selected arm is local. The challenge is a tradeoff for agents between pulling a local arm with observable feedback or pulling external arms without feedback and relying on others' observations that occur at different rates. We propose `AAE-LCB`, a two-stage algorithm that prioritizes pulling local arms following an active arm elimination policy and switches to other arms only if all local arms are dominated by some external arms. We analyze the regret of `AAE-LCB` and show it matches the regret lower bound up to a small factor.

## 1 Introduction

Multi-armed bandits (MABs) [16, 50] is a remarkably successful online learning framework that has been extensively studied since the 1950s [47]. It has a broad range of applications including datacenter optimization, web advertising, and recommender systems [50, 31, 39, 15]. In the basic MAB problem, a learner repeatedly chooses an action (pulls an arm) in each round, and collects (and observes) the reward associated with the selected arm, but not rewards associated with the unselected arms. The goal of the learner is to maximize the long-term reward collected, and the performance metric is regret, which is the difference between the expected reward for the best arm and that received by the learner.

The distributed/multi-agent/multiplayer MAB problem is an extension of the basic MAB that has been studied extensively in different settings [43, 51, 11, 36, 33, 25, 45, 53, 48]. This problem is motivated by several applications such as (1) large-scale learning systems [19] in domains such as finance or recommendation systems; (2) cooperative search by multiple robots [41, 32]; (3) the application of wireless cognitive radio [17, 43, 42, 9]; and (4) distributed learning in distributed systems (e.g., a set of IoT devices learning about an underlying environment) [46, 23, 5, 12, 54].

35th Conference on Neural Information Processing Systems (NeurIPS 2021).

Most prior work on the multi-agent MAB problem assumes that agents have full access to all arms, and hence they solve the same MAB problem with the goal of minimizing the aggregate regret of the agents either in a *competition* setting [3, 14, 17, 53, 11, 13, 43, 42, 9] where agents receive degraded or no rewards when pulling the same arm, or in a *collaboration/cooperation* setting [45, 53, 38, 37, 36, 48, 44], where agents receive independent rewards when they pull the same arm, and agents can communicate to improve their learning performance. In this work, we focus on the cooperative version of this problem, which has been extensively studied in recent years under different settings. In a thread of work, the agents are considered to be connected together over a communication graph [45, 53, 38, 37, 10, 24]. Among them, [53] presents a more practical model in which agents are limited in their available communication capacity with others, and also develops the state-of-the-art algorithm which achieves an asymptotically optimal regret. The above basic models have been extended to several other settings such as pure-exploration through cooperation among agents for identifying the best stochastic arm [29], cooperative multi-agent bandits with heavy tails [25], cooperative kernelised contextual bandits [27], linear bandits with safety constraints [2], and the case with both honest and malicious agents [52]. In [10], the problem is extended to the case where the underlying topology is time-varying. In another trend, the cooperative bandits have been studied over social networks [48, 51, 36]. Last, in a very recent trend, the stochastic bandits have extended to the federated setting, in which the learning task is distributed among multiple agents [56, 49, 26].

In this paper, we study a heterogeneous version of the cooperative multi-agent stochastic bandit problem with a set $\mathcal{A} = \{1, \ldots, M\}$ of agents and a set $\mathcal{K} = \{1, \ldots, K\}$ of arms. Agent $j \in \mathcal{A}$ has access to a subset $\mathcal{K}_j \subseteq \mathcal{K}$ of arms. We refer to arms in $\mathcal{K}_j$ as *local* and the remaining arms in $\mathcal{K} \setminus \mathcal{K}_j$ as *external* arms. Agents come with different learning capabilities and can pull an arm every $1/\theta_j$ rounds, with $0 < \theta_j \leq 1$ as the action rate of agent $j$. The goal is to collaboratively find the optimal arm in $\mathcal{K}$. An agent can pull any arm and receives a reward; however, it *observes* the reward only when it pulls a local arm. It instantaneously forwards this observation to all other agents. We call this setup *Feedback-constrained Cooperative Multi-agent MAB* (FC-CMA2B) and formally define it in Section 2.

The FC-CMA2B setting arises naturally in large-scale learning systems that are often geographically distributed for domains such as web search, finance, and recommendation systems. In such large-scale systems, it is usually impossible to run a single bandit algorithm. Instead, a distributed set of agents work to solve the problem, each with access to a subset of the action space. However, the entire learning task is integrated and the goal is to find the best global action, e.g., the most relevant search result in a web search application, or the best possible video recommendation in the YouTube recommendation system among millions of possible recommendations. In the above scenario, learning among different agents may be *asynchronous* in the sense that each agent has its own action (decision making) rate, which is captured by FC-CMA2B. In Appendix A, we provide a few concrete application scenarios that could be captured by FC-CMA2B.

The algorithmic challenge in FC-CMA2B originates from a nontrivial tradeoff that an agent must make between pulling a local arm with an observable reward, and pulling an external arm with the expectation of a larger but not observable reward. One might simply run a cooperative UCB algorithm on each agent without any distinction between local and external arms and pull the arm with the highest confidence interval. Then, if the selected arm is local, the agent can broadcast its observation to others. However, this strategy suffers a poor regret. The reason is intuitive. Consider a case where a suboptimal arm is only observable by a slow agent, i.e., an agent with a very small action rate $\theta_j$. In such a case, the upper confidence bound of the suboptimal arm will remain large due to the limited observations available only to the slow agent. Since the indexing policy of cooperative UCB suggests pulling the arm with the highest upper confidence bound, then, the suboptimal arm will be continuously pulled by everyone, resulting in a poor regret.

We note that [55] also considers a cooperative bandit setting with heterogeneous agents with a subset of arms and different action rates. However, in [55], the goal of each agent is to find the local optimal arm, while in this work, the goal of each agent is to find the global optimal arm. This difference leads to substantially different challenges in the algorithm design and analysis. More specifically, in our work, a foundational challenge is to find an effective cooperative strategy to resolve a dilemma between pulling local vs. external arms. This is not the case in [55] since the goal is to find the best local action.

**Contributions.** This paper makes the following contributions.

*Algorithm design using a two-stage learning strategy.* To tackle the new dilemma in `FC-CMA2B` for pulling either local or external arms, a proper bandit algorithm should be able to collect sufficient reward information, which is possible only by pulling local arms. To achieve this, our high-level idea is to distinguish between local and external arms and prioritize the local arms and only switch to external arms when they are clearly better. We implement this idea by proposing `AAE-LCB`, a two-stage learning strategy, which in the first stage selects local arms based on a carefully-designed *active arm elimination* (`AAE`) policy, and in the second stage selects external arms based on a *lower confidence bound* (LCB) policy.

More specifically, in the *first stage* of `AAE-LCB`, each agent pulls only from a dynamically-constructed candidate set of local arms. Then through an `AAE` process, a local arm is removed from the candidate set when its confidence interval falls below that of at least one other (local or external) arm. With this strategy, `AAE-LCB` makes a sufficient number of observations of local arms, and resolves issues encountered by the cooperative UCB. For an agent whose local arms do not include the optimal arm, the elimination process continues until all local arms are eliminated, and when so, the algorithm enters a *second stage*, where the agent only pulls external arms.

In the second stage, the agent totally relies on external observations that may occur at different rates. Differences in observation rates complicates decision making regarding choice of external arm such that the agent has to balance between well observed arms, and those that are not well observed but likely to be optimal. To tackle this tradeoff, `AAE-LCB` pulls an external arm with the largest LCB, and the intuition is that an arm with greater LCB is both better observed and more likely to be optimal. Note that selecting based on largest UCB may lead to pulling external arms whose upper confidence bounds are large due to a lack of samples.

*Regret lower and upper bounds.* We analyze the regret of `AAE-LCB` and show that it achieves a regret of $O(\sum_{i:\Delta_i>0} \Theta_i/\Theta_{i^*} \times K \log T/\Delta_i)$, where $\Theta_i$ is the action rate of arm $i$ defined as the sum of the action rates of agents containing arm $i$ and $i^*$ is the optimal arm. We further establish a regret lower bound for `FC-CMA2B` and shows `AAE-LCB` is regret optimal up to a small factor. For comparison, we investigate the regret of a baseline algorithm, `AAE-AAE`, which pulls external arms based on another `AAE` process and show that it exhibits poor performance in the presence of agents with low action rates. Specifically, we show that `AAE-AAE` suffers a regret of $\Omega(\Theta/\Theta_{\min} \log T)$, where $\Theta = \sum_{j\in\mathcal{A}} \theta_j$ and $\Theta_{\min} = \min_{i\in\mathcal{K}} \sum_{j:i\in\mathcal{K}_j} \theta_j$ are the aggregate action rate of all agents, and the *minimum* aggregate action rate of agents containing any arm $i$, respectively. This result shows that regret can be arbitrarily bad when $\Theta_{\min}$, i.e., the minimum observation rate among all arms (either optimal or suboptimal), is small. Hence, instead of regret depending on the observation rate of the least observable (and possibly suboptimal) arms, the regret of `AAE-LCB` depends on that of the optimal arm.

*Numerical results.* Through brief numerical experiments, we verify our theoretical observations and show that the cooperative extension of `UCB` and `AAE-AAE` are either unable to gain sufficient observations for the global optimal arm or vulnerable to the action rate of slow agents with suboptimal arms, while `AAE-LCB` is robust to both issues and achieves much better performance.

## 2 Problem Setup

**Feedback-constrained Cooperative Multi-agent MAB** (`FC-CMA2B`)**.** We consider a multi-agent stochastic bandit setting with a set $\mathcal{A} = \{1, \ldots, M\}$ of independent agents existing over the entire time period, and a set $\mathcal{K} = \{1, \ldots, K\}$ of arms. Associated with arms are mutually independent sequences of i.i.d. Bernoulli rewards with mean $0 \leq \mu(i) \leq 1$, for arm $i \in \mathcal{K}$. Agent $j \in \mathcal{A}$ has full access to a subset $\mathcal{K}_j \subseteq \mathcal{K}, K_j = |\mathcal{K}_j|$, of arms. We refer to $\mathcal{K}_j$ and $\mathcal{K} \setminus \mathcal{K}_j$ as the sets of local and external arms of agent $j$, respectively. Agents are allowed to pull and receive a reward from any arm from $\mathcal{K}$, however, they only receive observations from local arms. That is to say, if the selected arm is external, the agent does not observe the reward even though the reward is allocated. Last, we make no assumption regarding overlaps among sets $\mathcal{K}_j$.

In addition to heterogeneity in access to arms, agents are also heterogeneous in their decision making capabilities. Specifically, considering the decision rounds $\{1, \ldots, T\}$, agent $j$ is able to pull an arm every $\omega_j \in \mathbb{N}^+$ rounds, i.e., decision rounds for agent $j$ are $t = \omega_j, 2\omega_j, \ldots, N_j\omega_j$, where $N_j = \lfloor T/\omega_j \rfloor$. Parameter $\omega_j$ represents the *inter-round gap* of agent $j$. For simplicity of analysis,

we define $\theta_j := 1/\omega_j$ as the *action rate* of agent $j$. Intuitively, the larger $\theta_j$, the faster the agent $j$ is in pulling arms.

The goal of each agent is to learn the global optimal arm. The regret for agent $j$ in `FC-CMA2B` is

$$R_T^j := \mu(i^*)N_j - \sum_{t \in \{k\omega_j : k=1,\ldots,N_j\}} x_t(I_t^j),$$

where $i^*$ is the optimal arm in $\mathcal{K}$, and $I_t^j \in \mathcal{K}$ is the pulled arm by agent $j$ at time $t$. Agent $j$ receives the reward $x_t(I_t^j)$ whose value is observable only if $I_t^j \in \mathcal{K}_j$. The ultimate goal is to minimize the aggregate regret of the agents, i.e., $R_T = \sum_{j \in \mathcal{A}} R_T^j$.

We note that when two agents select the same arm, they collect stochastically independent rewards, and there is no reward degradation such as occurs in competition settings. We also consider full and truthful cooperation among agents, i.e., after committing to one decision and receiving a reward from the pulled arm, an agent broadcasts observations to others at no cost and no delay. In the supplementary material, we further consider a more general model with communication delays among agents.

**Additional notations and terminologies.** To facilitate our algorithm design and analysis, we introduce the following notations. Let $\mathcal{A}_i$ denote the set of agents whose local arm sets include arm $i$,

$$\mathcal{A}_i := \{j \in \mathcal{A} : i \in \mathcal{K}_j\}, \quad \forall i \in \mathcal{K}.$$

By $\Delta(i', i)$, we denote the difference in mean rewards of arms $i'$ and $i$, i.e., $\Delta(i', i) := \mu(i') - \mu(i)$. When $i' = i^*$, we rewrite $\Delta(i^*, i)$ as $\Delta_i$, which is known as the suboptimality gap in the basic bandit problem. Last, we define $\Theta$ and $\Theta_i$, $i \in \mathcal{K}$, as follows

$$\Theta := \sum_{j \in \mathcal{A}} \theta_j, \quad \text{and} \quad \Theta_i := \sum_{j \in \mathcal{A}_i} \theta_j,$$

where $\Theta_i$ is the action rate of arm $i$ across all agents in $\mathcal{A}_i$. Intuitively, the larger $\Theta_i$ is, the higher the rate at which arm $i$ can be pulled by agents in $\mathcal{A}_i$. The action rate $\Theta_i$ plays a key role in characterizing the regret bounds.

## 3 Algorithm Design

Each agent in the `FC-CMA2B` setting has to resolve a tradeoff between two categories of arms: (i) local arms with observable rewards, and (ii) external arms whose rewards are not observable but may be better than local arms. This heterogeneity in information feedback motivates our two-stage learning algorithm, `AAE-LCB`. In the first stage, agents running `AAE-LCB` are focused on sampling local arms to collect sufficient information that is useful for all agents in finding the global optimal arm. And when with high probability, an agent is confident that its local set does not contain the optimal arm, it moves to the second stage by pulling external arms based on others' observations.

### 3.1 Stage 1: Selecting a local arm

In order to converge to the global optimal arm, agents must collect sufficient observations in the first stage. Toward this, our idea is to prioritize pulling local arms as much as possible and only switch to external arms when the agent finds an external arm that is *sufficiently better* than all its local arms with high confidence. To implement this idea, we extend the active arm elimination (AAE) algorithm [28, 4] to capture the additional issues in `FC-CMA2B`. Similar to the basic `AAE` algorithm, in our algorithms, we start off by constructing a dynamic candidate set of local arms and gradually eliminating arms from it. The elimination process is similar to the basic `AAE`, which eliminates an arm when its confidence interval falls below that of at least one other arm. However, the difference is that while the candidate set includes local arms, the criteria for eliminating an arm is based on a comparison between both local and external arms.

Specifically, we define the width of the confidence interval for arm $i$ and agent $j$ at time $t$ as

$$\mathtt{cint}(i, j, t) := \sqrt{\frac{\alpha \log \delta_t^{-1}}{2\hat{n}_t^j(i)}}, \tag{1}$$

---

**Algorithm 1** AAE-LCB: A Cooperative Bandit Algorithm for Agent $j$ in the FC-CMA2B setting

---

1: **Initialization:** $\hat{n}^j(i) = 0$, $\hat{\mu}(i)$, $i \in \mathcal{K}$, $\mathcal{C}_j = \mathcal{K}_j$, $\alpha > 2$, $\delta_t$.
2: **for** each decision round $t$ **do**
3:     **for** each new received $x_\tau(i)$, $i \in \mathcal{K}$, $\tau < t$, **do**
4:         Execute line (8)-(10) for arm $i$
5:     **end for**
6:     **if** $\mathcal{C}_j \neq \emptyset$ **then**            ▷ *Stage 1: Pulling local arms through an* AAE *process*
7:         Pull arm $I_t^j$ with the least no. of observations from $\mathcal{C}_j$
8:         Increase counter for the selected arm by 1: $\hat{n}^j(I_t) \leftarrow \hat{n}^j(I_t) + 1$
9:         Update estimate of the mean value of the selected arm $\hat{\mu}(I_t)$
10:        Reconstruct candidate set based on updated values of $\hat{n}^j(I_t)$ and $\hat{\mu}(I_t^j)$ as in Equation (2)
11:        Broadcast $x_t(I_t^j)$ to other agents
12:     **else**                  ▷ *Stage 2: Pulling external arms through a* LCB *process*
13:         Select an external arm with the largest lower confidence bound as in Equation (3)
14:     **end if**
15: **end for**

---

where $\hat{n}_t^j(i)$ is the total number of observations (either local or received from other agents) of arm $i$ available to agent $j$ by time $t$ (observations made in time slots from 1 to $t-1$). Also, $\delta_t > 0$ and $\alpha > 2$ are parameters of the confidence interval. The removal of arm $i$ from agent $j$'s candidate set at time $t$ with empirical mean of $\hat{\mu}(i, \hat{n}_t^j(i))$ occurs if there is another arm $i'$, such that the difference in empirical means of $i$ and $i'$ is larger than the widths of the confidence intervals, i.e., $\hat{\mu}(i', \hat{n}_t^j(i')) - \hat{\mu}(i, \hat{n}_t^j(i)) > \texttt{cint}(i', j, t) + \texttt{cint}(i, j, t)$. This means that with high probability, arm $i$ is not optimal. Note that $i'$ could be either a local or an external arm. More formally, agent $j$ constructs the following dynamic candidate set $\mathcal{C}_{j,t}$ at time $t$ during its learning process.

$$\mathcal{C}_{j,t} = \left\{ i \in \mathcal{K}_j : \hat{n}_t^j(i) = 0 \text{ or } \hat{\mu}(i, \hat{n}_t^j(i)) - \hat{\mu}(i', \hat{n}_t^j(i')) \leq \texttt{cint}(i, j, t) + \texttt{cint}(i', j, t), \forall i' \in \mathcal{K} \right\}. \tag{2}$$

Note that in the above construction the local candidate set of agent $j$ includes all local arms not yet observed and all others not dominated by any other local or external arm. If the local candidate set is not empty, an agent always prioritize selecting local arms and, following basic AAE, pulls a local arm in the candidate set with the least number of observations, and then updates the empirical mean values and broadcasts the observation to others as in Lines 8-10 of Algorithm 1. We also note that the size of the dynamic candidate set is not necessarily decreasing and it is possible that after pulling some external arms, some local arms are put back into the local candidate set. Hence, in run time, fluctuations between pulling local and external arms might be possible.

### 3.2 Stage 2: Selecting an external arm

As Stage 1 of the learning process moves forward, it is possible that the local candidate set of an agent will become empty, i.e., an external arm eventually dominates all local arms. In this situation, the question becomes how to pick an external arm. The answer to this question is critical in designing cooperative algorithms with low regret.

Different action rates among different agents can lead arms to suffer different observation limitations that can invalidate the selection indices commonly used in the bandit algorithms. Specifically, when a suboptimal arm is only accessible to slow agents, estimates of the empirical mean reward of this arm will be updated more slowly than those of other arms pulled by faster agents, and thus contain more errors. In this way, those algorithms, which are unaware of confidence of estimates, may be misled into selecting a suboptimal arm with low-level confidence in the FC-CMA2B setting. For example, "slow arms", i.e., those arms that are only in the local sets of slow agents, have much looser confidence interval and a much larger upper confidence bound than others. As a result, fast agents running a cooperative version of AAE or UCB-based algorithms that pull external arms based on estimated upper confidence bounds may be misled into continuously selecting a suboptimal "slow" arm because its upper confidence bound is large due to insufficient number of observations. To address this, we propose to follow a Lower Confidence Bound (LCB) policy, which selects the external arm with the largest lower confidence bound, i.e.,

$$I_t^j = \arg\max_{i \in \mathcal{K}/\mathcal{K}_j} \hat{\mu}(i, \hat{n}_t^j(i)) - \texttt{cint}(i, j, t). \tag{3}$$

An important observation on LCB is that the lower confidence bound of an arm is large only if it is well observed and hence is likely to be the optimal arm. Indeed this is not the case if the selection policy is based on the largest upper confidence bound, since external arms with low observation might have large upper confidence bound.

### 3.3 Baseline Algorithms: `CO-UCB` and `AAE-AAE`

To show the advantages of `AAE-LCB`, we introduce two baseline algorithms. The first one is `CO-UCB`, a cooperative version of `UCB` that the agents do not distinguish between local and external arms. In other words, each agent running `CO-UCB`, pulls the arm with the largest upper confidence bound and, if the information is observable, broadcasts it to all other agents. Further, each agent will updates its confidence intervals not only based on its local observations, but also, based on the information received from other agents.

The second algorithm is `AAE-AAE`, which has a similar two-stage structure to prioritize pulling local arms as `AAE-LCB`, but adapts another layer of active arm elimination for pulling external arms. In the second stage, i.e., when the local set is emptied, `AAE-AAE` randomly pulls an external arm whose confidence interval have overlaps with others' and dynamically reconstructs the external candidate set similar to the process in the first stage, but for external arms. Because external arms with few observations has loose confidence interval, they satisfy the criteria to be in the candidate set, hence, the agents running `AAE-AAE` may repeatedly pull them.

In Section 5, we numerically compare the performance of both baseline algorithms with `AAE-LCB` and show the poor performance of `CO-UCB` due to the same treatment for local and external arms, and the improvement of `AAE-LCB` over `AAE-AAE` due to better treatment of external arms. In addition, in the next section, we theoretically compare the regret of `AAE-LCB` and `AAE-AAE`.

## 4 Regret Results

Two main regret results are highlighted in the following two theorems. We first introduce some terminology to facilitate the presentation of the results. Consider an algorithm $\pi$ running on agents in FC-CMA2B. We say that $\pi$ is *consistent*, if its regret satisfies $\mathbb{E}[R_T(\pi)] = O((T\Theta)^\sigma)$ as $T \to +\infty$ for any $\sigma > 0$, and for any set of Bernoulli reward distributions. Further, let $\text{KL}(\mu_i, \mu_i + \Delta_i)$ refer to the the Kullback-Leibler divergence between a Bernoulli of parameter $\mu_i$ and $\mu_i + \Delta_i$.

**Theorem 1 (Asymptotic Regret Lower Bound for `FC-CMA2B`)** *For any consistent algorithm $\pi$ and any $0 < \sigma < 1$, its expected regret satisfies*

$$\liminf_{T \to +\infty, \Theta/\Theta_{i^*} \to +\infty} \frac{\mathbb{E}[R_T(\pi)]}{(\Theta/\Theta_{i^*})^\sigma \log(T\Theta)} = \Omega\Big( \sum_{i:\Delta_i>0} \frac{\Delta_i}{\text{KL}(\mu_i, \mu_i + \Delta_i)} \Big).$$

The proof for the lower bound consists of two steps. In the first one, we prove that under any given agent action rates, any algorithm suffers a regret of $\Omega(\log(T\Theta))$ for large $T$. This proof uses techniques commonly used to prove lower bounds for stochastic bandits. In the second step, we prove that, with large enough $T$ and $\Theta/\Theta_{i^*}$, no algorithm can have a dependency on $(\Theta/\Theta_{i^*})^\sigma$ for $0 < \sigma < 1$. This is proved by contradiction. The details are given in the supplementary material.

**Theorem 2 (Expected Regret for `AAE-LCB`)** *Define $\delta := \min_{l,j} \delta_{l/\theta_j}$. The expected regret of* `AAE-LCB` *with parameter $\alpha > 2$ has the following upper bound*

$$\mathbb{E}[R_T] \leq \sum_{i:\Delta_i>0} \max\left\{ \frac{4\alpha \log \delta^{-1}}{\Delta_i}\left(2 + \frac{K\Theta_i}{\Theta_{i^*}}\right), \frac{12\Theta_i \alpha K \log \delta^{-1}}{\Delta_i \Theta_{i^*}} \right\}$$
$$+ 2\left(1 + \frac{\Theta_i}{\Theta_{i^*}}\right) \sum_{j \in \mathcal{A}} \sum_{l=1}^{N_j} \sum_{i \in \mathcal{K}_j} \frac{l\Theta_i}{\theta_j} \delta_{l/\theta_j}^\alpha.$$

**Corollary 1** *With $\delta_t = 1/t$ and $\alpha > 2$, we have the following regret for the* `AAE-LCB` *algorithm.*

$$\mathbb{E}[R_T] = O\left( \sum_{i \in \mathcal{K}:\Delta_i>0} \frac{K\Theta_i \log T}{\Theta_{i^*} \Delta_i} \right).$$

The prove is given in Section 4.1. From Corollary 1 and Theorem 1, we observe that the asymptotic regret of the proposed `AAE-LCB` algorithm matches the lower bound up to a small factor $O(K(\Theta/\Theta_{i^*})^{1-\sigma})$, where $\sigma$ is arbitrarily close to 1.

We also compare the regret of `AAE-LCB` to that of `AAE-AAE`. With $\delta_t = 1/T$, and $\alpha > 4$, the regret of `AAE-AAE` could be characterized as

$$\mathbb{E}[R_T] = \Omega\left(\frac{\Theta}{\Theta_{\min}}\log T\right), \tag{4}$$

where $\Theta_{\min} := \min_i \Theta_i$. The detailed derivation of the the above regret is in the supplementary materials. The regret of `AAE-AAE` in (4) strongly depends on $\Theta_{\min}$, i.e., the smallest aggregate action rate among all arms. Hence, as $\Theta_{\min}$ goes to zero, the regret of `AAE-AAE` significantly degrades. The regret of `AAE-LCB` shown in Corollary 1, however, depends only on the action rate of the optimal arm. Hence, `AAE-LCB` outperforms `AAE-AAE`, in the sense that its performance is independent of the slowest arm.

## 4.1 A Proof for Theorem 2

There are two contributions to the regret of `AAE-LCB`: one due to pulling suboptimal local arms, and the other one due to pulling suboptimal external arms. To prove Theorem 2, we analyze these two contributions. Note that, observations are generated whenever agents pull local arms. Thus, the first contribution to regret can be upper bounded by analyzing the relationship between the confidence interval and the number of observations obtained from each arm. By analyzing the algorithm rules of `AAE-AAE`, we upper bound the expected number of observations required from suboptimal arms to identify the optimal arm if it lies in the local set, as well as the first contribution to regret. Regarding the second contribution to regret, the main difficulty comes form the heterogeneity of action rates associated with the arms. To upper bound this contribution to regret, we first upper bound the time period needed for the agents containing the optimal arm in their local sets generate enough observations such that the lower confidence bound of the optimal arm is higher than those of any others. With this, we can prove the the final result by upper bounding the number of times that suboptimal external arms need to be selected in the above proved time period.

Now, we proceed to formally prove the regret result. First, we categorize decisions made by the agents into Type-I and Type-II decisions. Type-I corresponds to the decisions of an agent when the actual mean values of local arms lie in the confidence intervals calculated by the agent, otherwise, Type-II decision happens, i.e., the actual mean value of some local arm is not within the calculated confidence interval. Specifically, when agent $j$ makes a Type-I decision at time $t$, we have

$$\mu(i) \in \left[\hat{\mu}\left(i, \hat{n}_t^j(i)\right) - \texttt{cint}(i, j, t), \hat{\mu}\left(i, \hat{n}_t^j(i)\right) + \texttt{cint}(i, j, t)\right], \quad i \in \mathcal{K}_j.$$

With Type-I decisions, an agent can keep the local optimal arm in its candidate set and eventually converges its decisions to the local optimal arm. The following Lemma, whose proof is given in the supplementary material, provides the probability that a Type-I decision happens at a particular decision round.

**Lemma 1** *Agent $j$ running `AAE-AAE` with $\alpha > 2$ makes a Type-I decision, in round $t = l/\theta_j$, with a probability of at least $1 - 2\sum_{i \in \mathcal{K}_j} \frac{l\Theta_i}{\theta_j}\delta_{l/\theta_j}^\alpha$.*

In the `AAE-LCB` algorithm, a suboptimal arm pulled by agent $j$ lies either in the candidate set or is the one with the largest lower confidence bound among all arms. Thus, we split the regret analysis into two cases: *Case I: local arm selection*, where a suboptimal decision is made during the arm elimination period for arms lying in the local set; and *Case II: external arm selection*, where a suboptimal decision is made based on the lower confidence bound when selecting external arms. In the following, we analyze the regret in each case separately.

*Case I: regret due to local arm selection:* For a suboptimal arm $i$, we upper bound the total number of pulling times by agents in $\mathcal{A}_i$, which is actually equal to $\hat{n}_t^j(i)$. First, we focus on the cases that the algorithm makes a Type-I decision at time $t$, i.e., the mean value of any arm lies in its confidence interval calculated by agent $j$. Then, at time $t$, if agent $j$ in $\mathcal{A}_i$ selects arm $i$, we have

$$2\texttt{cint}(i, j, t) + 2\texttt{cint}(i^*, j, t) \geq \Delta_i. \tag{5}$$

Otherwise, we have

$$\hat{\mu}(i, \hat{n}_t^j(i)) + \mathtt{cint}(i, j, t) \leq \mu_i + 2\mathtt{cint}(i, j, t) < \mu_i + \Delta_i - \mathtt{cint}(i^*, j, t)$$
$$= \mu_{i^*} - 2\mathtt{cint}(i^*, j, t) \leq \hat{\mu}(i^*, \hat{n}_t^j(i^*)) - \mathtt{cint}(i^*, j, t),$$

implying that arm $i$ is strictly dominated by $i^*$, hence, $i$ can not be selected by agent $j$, contradicting the assumption that $i$ is selected by $j$. It follows from Equation (5) that

$$\max\{2\mathtt{cint}(i, j, t), 2\mathtt{cint}(i^*, j, t)\} \geq \frac{\Delta_i}{2},$$

and by the definition of $\mathtt{cint}(.)$ in Equation (1), we have

$$\min\left\{\hat{n}_t^j(i), \hat{n}_t^j(i^*)\right\} \leq \frac{8\alpha \log \delta_t^{-1}}{\Delta_i^2}. \tag{6}$$

Now, we focus on Type-II decisions. Let $Q$ denotes the number of Type-II decisions. By Lemma 1, we have

$$\mathbb{E}[Q] \leq 2 \sum_{j \in \mathcal{A}} \sum_{l=1}^{N_j} \sum_{i \in \mathcal{K}_j} \frac{l\Theta_i}{\theta_j} \delta_{l/\theta_j}^{\alpha}. \tag{7}$$

By combining Equations (6) and (7), we have

$$\min\left\{\mathbb{E}\left[\hat{n}_T^j(i)\right], \mathbb{E}\left[\hat{n}_T^j(i^*)\right]\right\} \leq \frac{8\alpha \log \delta^{-1}}{\Delta_i^2} + \mathbb{E}[Q]. \tag{8}$$

Then, we have

$$\mathbb{E}\left[\hat{n}_T^j(i^*)\right] \geq \frac{T\Theta_{i^*} - \mathbb{E}[Q]}{K} \geq \frac{\Theta_{i^*} n_T(i)}{\Theta_i K} - \frac{\mathbb{E}[Q]}{K}, \tag{9}$$

where the first inequality is based on the fact that the expected number of decision rounds with the optimal arm in the candidate set is at least $T\Theta_{i^*} - \mathbb{E}[Q]$, and the second one is based on the fact that $T \geq \hat{n}_T(i)/\Theta_i$. Combining the results in Equations (8) and (9), we get

$$\mathbb{E}[\hat{n}_T(i)] \leq \max\left\{\frac{8\alpha \log \delta^{-1}}{\Delta_i^2}, \frac{8\alpha K \Theta_i \log \delta^{-1}}{\Theta_{i^*} \Delta_i^2}\right\} + \left(1 + \frac{\Theta_i}{\Theta_{i^*}}\right) \mathbb{E}[Q]. \tag{10}$$

*Case II: regret due to external arm selection:* Now, we aim at upper bounding the expected number of selection times for a suboptimal arm $i$ by the agents outside set $\mathcal{A}_i$. Again, we assume that agent $j$ makes a Type-I decision at time slot $t$. Consider the case that $I_t^j = i$ and $i$ is not within $\mathcal{K}_j$. By algorithm rules, we have that arm $i$ has the largest lower confidence bound. We claim that

$$2\sqrt{2}\mathtt{cint}(i^*, j, t) \geq \Delta_i. \tag{11}$$

Otherwise, we have

$$\hat{\mu}(i, \hat{n}_t^j(i)) - \mathtt{cint}(i, j, t) \leq \mu(i) = \mu(i^*) - \Delta_i$$
$$< \hat{\mu}(i^*, \hat{n}_t^j(i^*)) + \mathtt{cint}(i^*, j, t) - 2\mathtt{cint}(i^*, j, t)$$
$$= \hat{\mu}(i^*, \hat{n}_t^j(i^*)) - \mathtt{cint}(i^*, j, t),$$

contradicting the rules of the algorithm. Thus, at time $t$, the selected arm $i$ satisfies

$$\Delta_i \leq 2\sqrt{\frac{\alpha K \log \delta_t^{-1}}{t\Theta_{i^*} - Q}}.$$

The above equation is obtained by replacing $\hat{n}_t^j(i^*)$ in equation $2\sqrt{2}\mathtt{cint}(i^*, j, t) \geq \Delta_i$ with $[t\Theta_{i^*} - Q]/K$, since $\hat{n}_t^j(i^*) \geq [t\Theta_{i^*} - Q]/K$.

For any agent $j$, the largest time slot when the agent makes a Type-I decision and a suboptimal arm $i$ lies in the candidate set is $4\frac{\alpha K \log \delta^{-1}}{\Delta_i^2 \Theta_{i^*}} + \frac{Q}{\Theta_{i^*}}$. Then, the regret spent on the arm $i$ in other agents is upper bounded by $4\frac{\alpha K \log \delta^{-1}}{\Delta_i} \frac{\Theta_i}{\Theta_{i^*}} + \frac{Q\Theta_i}{\Theta_{i^*}} + Q$.

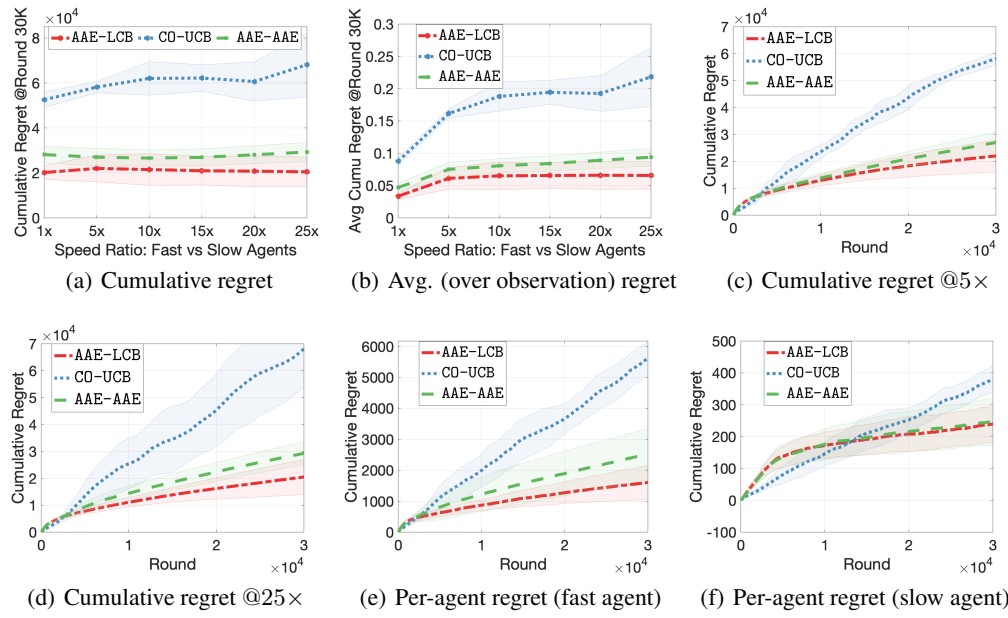

Figure 1: Regret of `AAE-LCB` vs. `AAE-AAE` vs. `CO-UCB` with two groups of "fast" and "slow" agents and varying action rate ratio between them. Notable observations: `AAE-LCB` outperforms `CO-UCB` significantly and outperforms `AAE-AAE` slightly. Comparing (c) and (d) shows that improvement of `AAE-LCB` over `CO-UCB` increases as the difference between action rates increases. Comparing (e) and (f) shows substantial regret degradation of `CO-UCB` in fast agent due to existence of slow agents.

Summing up the above two pieces of regret and the expected number of Type-II decisions yields

$$\mathbb{E}\left[R_T\right] \leq \sum_{i:\Delta_i>0} \max\left\{\frac{4\alpha\log\delta^{-1}}{\Delta_i}\left(2+\frac{K\Theta_i}{\Theta_{i^*}}\right), \frac{12\Theta_i\alpha K\log\delta^{-1}}{\Delta_i\Theta_{i^*}}\right\}$$
$$+2\left(1+\frac{\Theta_i}{\Theta_{i^*}}\right)\sum_{j\in\mathcal{A}}\sum_{l=1}^{N_j}\sum_{i\in\mathcal{K}_j}\frac{l\Theta_i}{\theta_j}\delta^\alpha_{l/\theta_j}.$$

This completes the proof.

## 5    Numerical Experiments

Our goal in this section is to numerically investigate the performance of `AAE-LCB` and compare it to that of `AAE-AAE` and `CO-UCB` (see Section 3.3), and show that `AAE-LCB` effectively resolves the challenge slow agents present, while neither `AAE-AAE` nor `CO-UCB` do so. More specifically, by comparing `AAE-LCB` with `CO-UCB`, our goal is to verify the importance of two-stage learning in the design of `AAE-LCB`, and by comparing `AAE-LCB` and `AAE-AAE`, our goal is to verify the importance of using LCB as the indexing policy for external arm selection.

**Experimental setup.**    We assume there are $K = 100$ arms with Bernoulli rewards with average rewards uniformly randomly taken from *Ad-Clicks* [1]. In our experiments, we report the cumulative regret after 30,000 rounds, which corresponds to the number of decision rounds of the fastest agent. All reported values are averaged over 20 independent trials. We have 20 agents, each with 12 arms selected from among a set of $K = 100$ arms into two categories of 10 "fast" agents each with action rate of 1, and 10 "slow" agents with varying action rates of less than 1. We compare the performances of `AAE-LCB` and `AAE-AAE` and `CO-UCB` that are introduced in Section 3.3.

**Experimental results.**    In Figures 1(a) and 1(b), we fix the action rate of the fast agents and vary the ratio between action rate of fast and slow agents from $5\times$ to $25\times$ with steps of 5. The results

show relatively sound performance of `AAE-LCB` as the speed ratio between fast and slow agents increases. However, the performance of `CO-UCB` decreases substantially as shown in Figure 1(b) for average per-agent regret, which is mainly due to treating local and external arms in the same way. This further verifies our intuitions for the need to distinguish between local and external arms as the main motivation for developing `AAE-LCB`. `AAE-AAE` and `AAE-LCB` exhibit similar performance showing the importance of prioritizing the local arms, however, `AAE-LCB` performs slightly better since it uses a better external arm selection policy than that of `AAE-AAE`. The evolution of cumulative regrets over time in Figures 1(c) and 1(d) and per-agent regret for fast and slow agents in Figures 1(e) and 1(f) also demonstrate that `AAE-LCB` outperform the alternatives.

Note that the above experimental scenarios are designed to demonstrate the "average-case" performance, since rewards of arms are generated according to real data traces and all arms are randomly allocated to fast and slow agents. However, another important consideration on the efficiency of a learning algorithm is its performance in the worst case. From our theoretical results in Section 4, baseline algorithms suffers severe performance degradation and our algorithm outperforms the baseline algorithms a lot only when the optimal arm lies in a fast agent and there exists some arm with few observations. Specifically, the regret of the `AAE-AAE` algorithm degrades with the smallest aggregate action rate of agents containing arm $i, i \in \mathcal{K}$, i.e. $\Theta_{\min}$, while that of `AAE-LCB` depends on the action

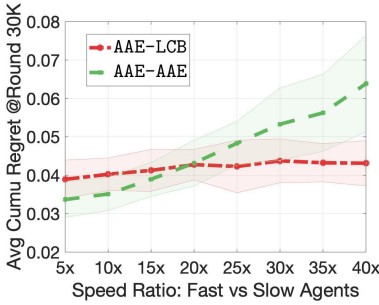

Figure 2: Performance comparison in the "worst" case

rate of the agent which the optimal arm lies in. To validate the efficiency of `AAE-LCB` in the worst case, we straightforwardly add a synthetic scenario and generate a special worst-case instance where the optimal arm only lies in fast agents. Specifically, we rank the arms in a descending order based on their reward means, and allocate a half of the arms with high reward means only to the fast agents and the other half only to the slow agents. The results in Figure 2 show as the ratio between the action rates of slow and fast agents increases, `AAE-LCB` outperforms `AAE-AAE` significantly.

## 6   Conclusion and Future Directions

In this paper, we proposed bandit algorithms with near-optimal regret for a cooperative stochastic bandit problem among multiple agents playing the same instance of the problem, each with limited access to the set of arms and with different decision-making capabilities.

A limitation of this work is that in the current result, there is a gap between the regret of `AAE-LCB` and the regret lower bound. We leave developing an algorithm with the optimal regret for the `FC-CMA2B` setting as an open problem. Additionally, a critical, yet practically relevant question is how to extend the algorithms to the case with communication delay and cost between agents. We address the communication delay in the supplementary material of the paper. Regarding communication cost, the question is how to design algorithms that can provide a tradeoff between the regret and communication costs. While this tradeoff has been studied extensively in recent works [18, 21, 29, 51], none of them consider this tradeoff in the presence of heterogeneous agents. This task is challenging since in each decision-making round, each agent should make a nontrivial decision on whether or not broadcast with other agents or a subset of agents. Another question arises by considering topological constraints for agents. In this work, we focus our analysis to a set of fully-connected agents. In practice, however, geographically distributed agents might have limited access to other agents over an underlying graph. This setting also has been studied in recent works such as [36, 48]; however, there is no work on the heterogeneous version of this setting. Last, we do not see any negative societal impacts of our work.

## Acknowledgments and Disclosure of Funding

We acknowledge the support from U.S. Army Research Laboratory Army Research Laboratory under Cooperative Agreement W911NF-17-2-0196 (IoBT CRA) and from U.S. Army Research Laboratory and the U.K. Ministry of Defence under Agreement W911NF-16-3-0001. We also acknowledge the support by NSF CAREER 2045641, CNS 1908298, 2102963, and CPS 2136199. The work of John C.S. Lui was supported in part by the GRF 14200420 and SRFS2122-4202.

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
