# A Motivating Application Examples

There are two new features in our multi-agent bandit setting: asynchronous sampling and partial access to arms. The asynchronous sampling has clear practical motivation due to the nature of multi-agent (or distributed) decision-making. In the following, we focus on the justification of practical relevance for the latter on partial access to arms. We highlight three different examples in the context of the online shortest path routing (OSPR), clinical trials, and crowdsourcing, which are classical applications of bandits. Indeed, our current model does not fully captures the following examples; instead, it provides initial modeling with interesting and nontrivial mathematical challenges that are addressed in this paper.

**OSPR.** In the basic version of OSPR, a bandit algorithm could be applied to select a path (arm) with minimum delay. Now consider an extended version of OSPR in a heterogeneous network that includes multiple virtual private networks (VPN) each represented by an agent (or a gateway, in networking terminology). In this scenario, the local arms of an agent represent the paths including the link in their VPN, and the paths in other VPNs are external arms. In the case of selecting a path that goes through some nodes in another VPN, the path information might not be observable to the rest of the network, which resides outside of the VPN, i.e., other agents. This scenario could be captured by our bandit setting in which an external arm (a path in another VPN) is selected and the reward is allocated (the delay), but it is not observable. On the other hand, when an agent selects a path in its local VPN, they can share the information with the representative agents of other VPNs to eventually find the globally shortest path in the entire network.

**Cooperative clinical trial.** A doctor in a clinic is required to offer a treatment plan for patients with some disease, while the clinic, which the doctor sits in, can only provide partial options due to lack of medical equipment. The goal of the doctor is to maximize his reputation or the number of treatment successes. In the medical decision-making setting, the plan for the disease corresponds to the arm, and the doctor in the clinic is an agent. If the doctor decides on a specific treatment option that is not offered in their clinic, the patient may not come back for follow-up appointments, and in this case, the patient's medical record or feedback might be missing. Thus, the treatment plans offered by the clinic are the local arms in `FC-CMA2B`. Cooperation among the doctors in different clinics helps to accelerate learning the best treatment plan, by sharing medical statistics.

**Crowdsourcing.** In crowdsourcing, workers are allowed to register to any agents, each of which is tailored for specific tasks. An agent maintains the profile of registered workers, with a reputation score (e.g., the success rate of finishing a job) updated with their performance in satisfying tasks. In real-world applications, workers are allowed to be anonymous to some agents with the aim to protect personal privacy, with only the reputation score revealed to the agent. For anonymous workers, the agent treats them as guests and no identity or profile can be tracked for them. Hence, the observed reward is useless. In crowdsourcing with multiple agents, anonymous workers correspond to external arms, and registered workers correspond to local arms. In this model, an agent selects anonymous workers only based on reputation scores by other agents. Implicitly, different agents cooperate by maintaining the common reputation score for registered workers.

It is worth noting that, to the best of our knowledge, `FC-CMA2B` is the first model that tackles a multi-agent bandit setting where agents have access to a subset of arms. The basic setting that we considered in this paper, however, could be extended to better capture more convincing practical applications. We highlight a few practically relevant, and (to our belief) feasible extensions. First, our model could be extended to the case that instead of exact information, a perturbed, yet useful observation is communicated between agents. This extension makes the setting much more interesting from a practical perspective since in this setting agents can share some limited information with others to incentivize them to use their local resources (arms). Another practical extension is adding the communication delay in the model as we tackled it in the Appendix E. One can consider communication costs in the model and the goal becomes to provide low regret algorithms with low communication costs.

## B  Extensive Literature Review

In addition to cooperative stochastic bandits, the cooperative nonstochastic bandits were introduced in [6] with the latest results in [20, 7, 30]. Specifically, [6] introduces the cooperation setting where agents share the distribution over actions without delay. They bound the average regret for the case that some of the agents are dishonest and behave in an arbitrary Byzantine manner. [20] studies the tradeoff between cooperation and delay, which is ruled by the underlying communication network topology, and proves a bound for the average regret. The authors in [7] prove an individual regret bound that holds simultaneously for all agents, solving an open problem left in [20].

Our work is different from all above related literature, since we consider heterogeneous agents each with access to a subset of arms and different decision capabilities. Besides, we consider fully-connected agents, while most of prior work on cooperative bandits considers an underlying graph connecting the agents. Extension of our result to the case with topology constraints and developing efficient communication protocol among agents are interesting future directions.

We note that asynchronous online learning is getting attention in recent works such as [19, 22]. Specifically, [19] considers a model, in which there is a network of agents, and in each round some of the agents are activated to make decisions. However, their model assumes full information feedback and ours is dedicated to the bandit setting. In the context of bandit setting, the asynchronous nature of our work is related to the category of sleeping bandits [40, 35, 34], in which some arms could be "sleeping" or "unavailable" at some rounds. In these models, typically the set of available arms changes without following any structure. In contrast, in our model, all arms are always available, but there is a structured limitation on the observability of feedback dictated by the local subset and action rate of agents. Note that the asynchronous nature our work considers is also different from the one in [14], which concerns the learning horizon of agents, i.e., whether an agent joins the game at the beginning, while we study the decision-making rates of agents.

Last, we note that in literature there is multi-agent bandits with competition. In this model, when an agent selects an arm, it collects the reward only if no other agent pulls the same arm. Also, in some cases, the reward of an arm will be degraded if it is being pulled by multiple agents. This model is motivated by the application of distributed channel selection in wireless cognitive radio systems. Several variants of this model have been studied in recent works, e.g., [3, 14, 17, 53, 11, 13, 43, 42, 9]. This line of work stands in clear contrast to our work, since we focus on the cooperative version of multi-agent bandit problems where rewards are independently collected across agents.

## C  Summary of Notations

We list all notations used in this paper in Table 1.

## D  Supplementary Proofs and Analysis

In the first subsection, we first provide details on the derivation of the confidence intervals in `AAE-LCB`, which will be used in our later proofs.

### D.1  Analysis of Confidence Intervals in `AAE`

Let $X_{i,s}$, $s = 1, 2, \ldots, n$, be the random variable by which the $s$-th observed reward for arm $i \in \mathcal{S}$ is generated. For any positive $a$, we have

$$
\begin{aligned}
\Pr\{\hat{\mu}_{i,n} \geq \mu_i + a\} &\leq \Pr\left\{\frac{1}{n}\sum_{s=1}^{n}[X_{i,s} - \mu_i] \geq a\right\} \\
&= \Pr\left\{\exp\left(\theta\sum_{s=1}^{n}[X_{i,s} - \mu_i]\right) \geq \exp\left(\theta a n\right)\right\} \\
&\leq \mathbb{E}\left\{\exp\left(\theta\sum_{s=1}^{n}[X_{i,s} - \mu_i] - \theta a n\right)\right\} \\
&= \exp(-n\theta a)\mathbb{E}\left\{\exp(\theta[X_{i,s} - \mu_i])\right\}^{n},
\end{aligned}
$$

Table 1: Summary of notations related to `FC-CMA2B`

| Notation | Description |
| :---: | :--- |
| $t$ | Index of time slot |
| $T$ | The number of time slots |
| $K$ | The number of arms |
| $M$ | The number of agents |
| $\mathcal{K}$ | Set of all arms |
| $\mathcal{A}$ | Set of agents |
| $\mathcal{K}_j$ | Local subset of arms for agent $j$ with observable reward |
| $\mathcal{A}_i$ | Set of agents containing arm $i$ |
| $\theta_j$ | Action rate of agent $j$ |
| $w_j$ | Gap of adjacent decision rounds of agent $j$ |
| $N_j$ | Total number of decisions made by agent $j$ |
| $\Theta_i$ | Aggregate action rate of agents containing arm $i$ |
| $\Theta$ | Aggregate action rate of all agents |
| $x_t(i)$ | Reward of arm $i$ at time slot $t$ |
| $\mu(i)$ | Mean reward of arm $i$ |
| $\Delta(i, i')$ | Difference of mean rewards between arm $i$ and $i'$ |
| $\Delta_i$ | Difference of mean rewards between the optimal arm and arm $i$ |
| $I_t^j$ | Chosen arm by agent $j$ at $t$ |
| $\delta_t$ | Algorithm parameter used in `AAE-LCB` and `AAE` |
| $\delta$ | The minimum value for $\delta_t$ over the entire time horizon |
| $n_t(i)$ | Total number of observations on arm $i$ up to $t$ |
| $\hat{n}_t^j(i)$ | Number of observations on arm $i$ available to agent $j$ up to $t$ |
| $\hat{\mu}(i, n)$ | Empirical reward mean of arm $i$ with $n$ observations |
| $\texttt{cint}(i, j, t)$ | Width of the confidence interval for arm $i$ and agent $j$ at time $t$ |
| $\mathcal{C}_{j,t}$ | Candidate set of agent $j$ defined in `AAE-LCB` and `AAE-AAE` |
| $R_T$ | Regret over $T$ time slots |
| $R_T^j$ | Individual regret of agent $j$ over time horizon $T$ |
| $d_j$ | The largest delay from any agent to agent $j$ |
| $D$ | The largest delay between any two agents |
| $\text{KL}(a, b)$ | The Kullback-Leibler divergence between a Bernoulli of parameter $a$ and $b$ |
| $\text{KL}(\mathbb{P}_1, \mathbb{P}_2)$ | The Kullback-Leibler divergence between two random distributions $\mathbb{P}_1$ and $\mathbb{P}_2$ |

where $\theta$ can be any positive. The inequality bases on Chebyshev's inequality.

Let $\phi(\theta) = \log \mathbb{E} \left\{ \exp(\theta[X_{i,s} - \mu_i]) \right\}$, we have

$$
\begin{aligned}
\Pr\left\{ \hat{\mu}_{i,n} \geq \mu_i + a \right\} &\leq \exp(-n\theta a) \exp(n\phi(\theta)) \\
&\leq \inf_\theta \exp(-n(a\theta - \phi(\theta)) \\
&= \exp(-n\phi^*(a)).
\end{aligned}
$$

where $\phi^*(a)$ is defined as $\sup_\theta (a\theta - \phi(\theta))$.

Replacing $a$ with $(\phi^*)^{-1} \left( \frac{\log(1/\delta)}{n} \right)$, we have

$$
\Pr\left\{ \hat{\mu}_{j,n} \geq \mu_j + a \right\} \leq \exp(-n\phi^*(a)) = \exp\left( -n\frac{\log(1/\delta)}{n} \right) = \delta.
$$

In summary,

$$
\Pr\left\{ \hat{\mu}_{j,n} \geq \mu_j + (\phi^*)^{-1} \left( \frac{\log(1/\delta)}{n} \right) \right\} \leq \delta.
$$

Similarly, we can derive the probability for $\hat{\mu}_{j,n} \leq \mu_j + (\phi^*)^{-1} \left( \frac{\log(1/\delta)}{n} \right)$.

For the bounded random variable $X_{i,s}$, we have

$$(\phi^*)^{-1}\left(\frac{\log(1/\delta)}{n}\right) = \sqrt{\frac{\log(1/\delta)}{2n}}.$$

To construct the confidence intervals, we set the confidence probability as $\delta = 1/n^\alpha$. Accordingly, the upper/lower confidence bound for the $i$-th arm is $\hat{\mu}_{i,s} + \sqrt{\frac{\alpha \log(n)}{2n}}$ and $\hat{\mu}_{i,s} - \sqrt{\frac{\alpha \log(n)}{2n}}$.

## D.2 A Proof for Lemma 1

Based on the results in D.1, for any arm $i$ with $n$ observations, we have

$$\Pr\left(\mu(i) > \hat{\mu}(i,n) + \sqrt{\frac{\alpha \log \delta^{-1}}{2n}}\right) \le \delta^\alpha.$$

Then, by applying the above inequality to the decision of agent $j$ at time $t = l/\theta_j$, we get (12).

$$\Pr\left(\mu(i) > \hat{\mu}\left(i, \hat{n}^j_{l/\theta_j}(i)\right) + \sqrt{\frac{\alpha \log \delta^{-1}_{l/\theta_j}}{2\hat{n}^j_{l/\theta_j}(i)}}\right)$$
$$\le \sum_{s=1}^{l\Theta_i/\theta_j} \Pr\left(\mu(i) > \hat{\mu}(i,s) + \sqrt{\frac{\alpha \log \delta^{-1}_{l/\theta_j}}{2s}}\right) \le \frac{l\Theta_i}{\theta_j}\delta^\alpha_{l/\theta_j}. \tag{12}$$

The above equation shows that the probability that the true mean value of arm $i$ is above the upper confidence bound in agent $j$ at time $l/\theta_j$ is not larger than $\frac{l\Theta_i}{\theta_j}\delta^\alpha_{l/\theta_j}$. Similarly, for the lower confidence interval we have

$$\Pr\left(\mu(i) < \hat{\mu}\left(i, \hat{n}^j_{l/\theta_j}(i)\right) - \sqrt{\frac{\alpha \log \delta^{-1}_{l/\theta_j}}{2\hat{n}^j_{l/\theta_j}(i)}}\right) \le \frac{l\Theta_i}{\theta_j}\delta^\alpha_{l/\theta_j}.$$

Thus, the probability that the mean value of any arm in $\mathcal{K}_j$ at time $l/\theta_j$ lies in the confidence interval is lower bounded by

$$1 - 2\sum_{i \in \mathcal{K}_j} \frac{l\Theta_i}{\theta_j}\delta^\alpha_{l/\theta_j}.$$

This completes the proof.

## D.3 A Proof for Theorem 1

Theorem 1 can be proved in two steps.

(1) In the first step, we prove the lower bound without considering the influence of $\Theta/\Theta_{i^*}$. To do that, we assume the action rate of each agent is a constant. The techniques for the proof of the lower bound in this case have been investigated extensively. For the completion of analysis, we provide the details as follows. Let us define $\mathcal{E}_K$ as the class of $K$-armed stochastic bandits where each arm has a Bernoulli reward distribution. Assume that policy $\pi$ is consistent over $\mathcal{E}_K$, i.e., for any bandit problem $\nu \in \mathcal{E}_K$ and any $\sigma' > 0$, whose regret satisfies

$$R_T(\pi, \nu) = O((T\Theta)^{\sigma'}), \text{ as } T \to +\infty.$$

Let $\nu = [P_1, P_2, \ldots, P_K]$ and $\nu' = [P'_1, P'_2, \ldots, P'_K]$ be two reward distributions such that $P_k = P'_k$ except for $k = i$. Specifically, we choose $P'_i = \mathcal{N}(\mu_i + \lambda)$ and $\lambda > \Delta_i$. For stochastic bandits, we have the following divergence decomposition equation (one can refer to [8] for more details).

$$\mathrm{KL}(\mathbb{P}_{\nu,\pi}, \mathbb{P}_{\nu',\pi}) = \mathbb{E}_{\nu,\pi}[n_T(i)]\,\mathrm{KL}(P_i, P'_i),$$

where $\mathbb{P}_{\nu,\pi}$ is the distribution of $T$-round action-reward histories induced by the interconnection between policy $\pi$ and the environment $\nu$, and $\text{KL}(\mathbb{P}_{\nu,\pi}, \mathbb{P}_{\nu',\pi})$ measures the relative entropy between $\mathbb{P}_{\nu,\pi}$ and $\mathbb{P}_{\nu',\pi}$.

In addition, from the high-probability Pinsker inequality, we have

$$\text{KL}(\mathbb{P}_{\nu,\pi}, \mathbb{P}_{\nu',\pi}) \geq \log \frac{1}{2\left(\mathbb{P}_{\nu,\pi}(A) + \mathbb{P}_{\nu',\pi}(A^c)\right)},$$

where $A$ is any event defined over $\mathbb{P}_{\nu,\pi}$ and $\mathbb{P}_{\nu',\pi}$. By definition, the regret of policy $\pi$ over $\nu$ and $\nu'$ satisfies

$$R_T(\nu, \pi) \geq \frac{T\Delta_i}{2} \mathbb{P}_{\nu,\pi}\left(n_T(i) \geq \frac{T\Theta}{2}\right),$$

and

$$R_T(\nu', \pi) \geq \frac{T(\lambda - \Delta_i)}{2} \mathbb{P}_{\nu',\pi}\left(n_T(i) < \frac{T\Theta}{2}\right).$$

The above equation bases on the fact that the suboptimality gaps in $\nu'$ is larger than $\lambda - \Delta_i$.

Concluding the above two equations and lower bounding $\Delta_i$ and $(\lambda - \Delta_i)/2$ by $\kappa(\Delta_i, \lambda) := \min\{\Delta_i, \lambda - \Delta_i\}/2$ yields

$$\mathbb{P}_{\nu,\pi}\left(n_T(i) \geq \frac{T\Theta}{2}\right) + \mathbb{P}_{\nu',\pi}\left(n_T(i) < \frac{T\Theta}{2}\right) \leq \frac{R_T(\nu, \pi) + R_T(\nu', \pi)}{\kappa(\Delta_i, \lambda)T}.$$

We have

$$\begin{aligned}
\text{KL}(P_i, P_i')\mathbb{E}_{\nu,\pi}\left[n_T(i)\right] &\geq \log\left(\frac{\kappa(\Delta_i, \lambda)}{2} \frac{T\Theta}{R_T(\nu, \pi) + R_T(\nu', \pi)}\right) \\
&= \log(\frac{\kappa(\Delta_i, \lambda)}{2}) + \log(T\Theta) - \log(R_T(\nu, \pi) + R_T(\nu', \pi)) \\
&\geq \log(\frac{\kappa(\Delta_i, \lambda)}{2}) + (1 - \sigma')\log(T\Theta) + C,
\end{aligned}$$

where $C$ is a constant. The last inequality is based on the assumption that the algorithm is consistent. Taking $\lambda = \Delta_i$, for large $T$, we can lower bound the regret of any consistent policy $\pi$ as follows:

$$\liminf_{T \to +\infty} \frac{R_T}{\log(T\Theta)} \geq \liminf_{T \to +\infty} \frac{\sum_i \mathbb{E}_{\nu,\pi}\left[n_T(i)\right]\Delta_i}{\log(T\Theta)} = O(\sum_i \frac{\Delta_i}{\text{KL}(P_i, P_i')}).$$

(2) In the second step, we proceed to prove that the regret lower bound has a further asymptotically linear dependency on $(\Theta/\Theta_{i^*})^\sigma$ for any $0 < \sigma < 1$. We prove this by contradiction: assume that $0 < \sigma'' < 1$ exists and the regret of some algorithm has an asymptotically linear dependency on $(\Theta/\Theta_{i^*})^{\sigma''}$. That is

$$\limsup_{T \to +\infty, \frac{\Theta}{\Theta_{i^*}} \to +\infty} \frac{R_T(\pi, \nu)}{(\Theta/\Theta_{i^*})^{\sigma''}\log(T\Theta)} = O(\sum_i \frac{\Delta_i}{\text{KL}(P_i, P_i')}). \tag{13}$$

By similar reasoning, we have

$$\text{KL}(\mathbb{P}_{\nu,\pi}, \mathbb{P}_{\nu',\pi}) \geq \log \frac{1}{2\left(\mathbb{P}_{\nu,\pi}(A) + \mathbb{P}_{\nu',\pi}(A^c)\right)}, \tag{14}$$

Redefine $A$ as event $n_T(i^*) < (T\Theta)/2$. Similarly, we have

$$R_T(\nu, \pi) \geq \frac{T\Delta_i}{2} \mathbb{P}_{\nu,\pi}\left(n_T(i^*) < \frac{T\Theta}{2}\right),$$

and

$$R_T(\nu', \pi) \geq \frac{T(\lambda - \Delta_i)}{2} \mathbb{P}_{\nu',\pi}\left(n_T(i^*) \geq \frac{T\Theta}{2}\right).$$

Then, it follows from Equation (14) that

$$\mathrm{KL}(P_i, P_i')\mathbb{E}_{\nu,\pi}\left[n_T(i^*)\right] \geq \log(\frac{\kappa(\Delta_i, \lambda)}{2}) + \log(T\Theta) - \log(R_T(\nu, \pi) + R_T(\nu', \pi))$$

$$\geq \log(\frac{\kappa(\Delta_i, \lambda)}{2}) + \log(T\Theta) - \sigma'' \log(\frac{\Theta}{\Theta_{i^*}}) - \log\log(T\Theta) + C$$

$$\geq \log(\frac{\kappa(\Delta_i, \lambda)}{2}) + (1 - \sigma'') \log(T\Theta) - \log\log(T\Theta) + C,$$

where the last inequality uses the fact that $\Theta_{i^*} > 1/T$.

And, when $\Theta/\Theta_{i^*} \to +\infty$, we will have

$$T\Theta_{i^*} < \frac{1}{\mathrm{KL}(P_i, P_i')}((1 - \sigma'') \log(T\Theta) - \log\log(T\Theta)).$$

Thus, the above inequality will not hold, since $T\Theta_{i^*} \geq \mathbb{E}_{\nu,\pi}\left[n_T(i^*)\right]$. This contradicts the consistency condition in Equation (13). That means, for given $T$, any algorithm incurs a linear regret with respect to $(\Theta/\Theta_{i^*})^\sigma$, for any $0 < \sigma < 1$, when $\Theta/\Theta_{i^*}$ is large enough. We conclude our results below.

$$\liminf_{T \to +\infty, \frac{\Theta}{\Theta_{i^*}} \to +\infty} \frac{R_T}{(\Theta/\Theta_{i^*})^\sigma \log(T\Theta)} = \Omega\left(\sum_i \frac{\Delta_i}{\mathrm{KL}(P_i, P_i')}\right), \text{ for any } 0 < \sigma < 1.$$

This completes the proof.

### D.4 The Regret Analysis of `AAE-AAE`

Last, we provide analysis on the regret result of the baseline algorithm `AAE-AAE` in Equation (4).

Consider a scenario where the local subset of each agent only contains one different arm. We assume there exists a slow agent with the smallest action rate containing a suboptimal arm $\tilde{i}$. Since there is no delay between agents, the empirical mean values and confidence intervals for arms by different agents is the same. In the following equation, we use the results in Appendix D.1 to calculate the probability that there exists some arm whose mean value is above its confidence interval of width $\frac{1}{2}\sqrt{\frac{\alpha \log T}{2\hat{n}_{l/\theta_j}^j(i)}}$, i.e., for any $j$, we have

$$\Pr\left(\exists i, j, l : \mu(i) > \hat{\mu}\left(i, \hat{n}_{l/\theta_j}^j(i)\right) + \frac{1}{2}\sqrt{\frac{\alpha \log T}{2\hat{n}_{l/\theta_j}^j(i)}}\right)$$

$$= \Pr\left(\exists i, l = 1, 2, \ldots, N_j : \mu(i) > \hat{\mu}\left(i, \hat{n}_{l/\theta_j}^j(i)\right) + \frac{1}{2}\sqrt{\frac{\alpha \log T}{2\hat{n}_{l/\theta_j}^j(i)}}\right)$$

$$\leq \sum_{i \in \mathcal{K}} \sum_{s=1}^{T} \Pr\left(\mu(i) > \hat{\mu}(i, s) + \frac{1}{2}\sqrt{\frac{\alpha \log T}{2s}}\right)$$

$$\leq \sum_{i \in \mathcal{K}} \frac{1}{T^{0.25\alpha}} \leq \frac{K}{T^{0.25\alpha - 1}}.$$

The first inequality uses the fact that there is at most $T$ observations for each arm since each agent only contains one arm. Thus, the probability that the mean values of all arms lie in the confidence interval of width $\frac{1}{2}\sqrt{\frac{\alpha \log T}{2\hat{n}_{l/\theta_j}^j(i)}}$ is at least $1 - 2K/(T^{0.25\alpha - 1})$.

We assume the mean values of all arms lie in the confidence interval of width $\frac{1}{2}\sqrt{\frac{\alpha \log T}{2\hat{n}_{l/\theta_j}^j(i)}}$. At any time slot $t$, arm $\tilde{i}$ will not be eliminated if

$$\frac{1}{2}\sqrt{\frac{\alpha \log T}{2n_t(\tilde{i})}} \geq \Delta_{\tilde{i}}$$

Thus, under the above assumption, the number of observations needed to eliminate arm $\tilde{i}$ in the slow agent is at least

$$\frac{\alpha \log T}{8\Delta_{\tilde{i}}^2}.$$

Combining with the probability that the above assumption holds, we can provide a lower bound for the expected number of observations needed to eliminate arm $i$ in the slow agent as follows.

$$\frac{\alpha \log T}{8\Delta_{\tilde{i}}^2} \left(1 - \frac{2K}{T^{0.25\alpha - 1}}\right).$$

In other words, the expected number of mistakes made by the system can be as large as

$$\Omega \left(\frac{\Theta}{\Theta_{\min}} \frac{\alpha \log T}{8\Delta_{\tilde{i}}^2} \left(1 - \frac{2K}{T^{0.25\alpha - 1}}\right)\right).$$

Thus, the expected regret is at least $\Omega \left(\frac{\Theta}{\Theta_{\min}} \log T\right)$. This completes the proof.

## E  Extension to Multi-Agent Bandits with Delays

The goal of `FC-CMA2B` is to capture the heterogeneity in the action rates of different agents. The basic `FC-CMA2B` model introduced in this paper, however, can be extended to capture several additional practically relevant features, such as the cost of cooperation, delays in broadcasting observations, topology constraints, and malicious agents, each of which presents different additional challenges.

Here we extend our results to the case where there are communication delays between agents. Delays between agents are measured in units of decision rounds. We define $d_j$ to be the largest delay from any agent to agent $j$ with $D = \max_j d_j$. The delay between agents can be interpreted as the case in which agents are located on an underlying connected graph and the cooperation could be done by routing over a network. Then, assuming links with unit delay, the delay $d_j$ is the longest path from any agent to agent $j$ and $D$ is the diameter of the graph.

The `AAE-LCB` algorithm can be directly applied to the above case without any rule change. The regret analysis of `AAE-LCB` for `FC-CMA2B` with delays, however, needs to account for delays. The following result shows that `AAE-LCB` attains a regret with a linear dependency on the maximum delay parameter $D$.

**Theorem 3 (Expected Regret of `AAE-LCB` under `FC-CMA2B` with Delay)** *The expected regret of* `AAE-LCB` *has the following upper bound,*

$$\mathbb{E}\left[R_T\right] \le \sum_{i:\Delta_i > 0} \max \left\{\frac{4\alpha \log \delta^{-1}}{\Delta_i} \left(2 + \frac{K\Theta_i}{\Theta_{i^*}}\right) + F_i \Delta_i, \frac{12\Theta_i \alpha K \log \delta^{-1}}{\Delta_i \Theta_{i^*}} + KD\Theta_i \Delta_i\right\}$$

$$+ 2\left(1 + \frac{\Theta_i}{\Theta_{i^*}}\right) \sum_{j \in \mathcal{A}} \sum_{l=1}^{N_j} \sum_{i \in \mathcal{K}_j} \frac{l\Theta_i}{\theta_j} \delta_{l/\theta_j}^\alpha + D\Theta,$$

*where*

$$F_i := \sum_{j \in \mathcal{A}_i} \min \left\{d_j \theta_j, \frac{8\alpha \log \delta^{-1}}{\Delta_i^2}\right\}.$$

The proof of Theorem 3 follows the same logic flow as the proof of Theorem 2, and given later in this section.

Comparing the regret bounds in Theorems 2 and 3, the new one includes additional terms that are linear in the delays. Specifically, as $D$ increases to $T$, the regret will be linear, which is consistent with the fact that algorithms for `FC-CMA2B` suffers a linear regret when agents are totally separated.

We also briefly examine the impact of delay. Toward this, we consider three additional scenarios with average delays of 1000, 3000 and 5000 slots. Specifically, for average 1000 delay case, the mean

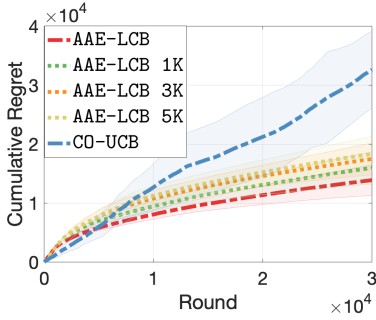

Figure 3: Regret of `AAE-LCB` with different delay values

delay between agent $j$ and $j'$, $d_{j,j'}$, is uniformly randomly picked from $[1000 - 10, 1000 + 10]$; and at each time slot, the exact delay is taken uniformly randomly from $[d_{j,j'} - 2, d_{j,j'} + 2]$. In Figure E, we report the evolution of cumulative regret of `AAE-LCB` and `CO-UCB` without any delay as well as `AAE-LCB` with delays. with $K = 100$, $M = 10$, 30 arms per agent. The results shows the regret of our algorithm for `FC-CMA2B` increases as the delay increases.

### E.1 A Proof for Theorem 3

The proof of Theorem 3 mainly follows that of Theorem 2 except incorporating delays in the analysis.

*Case I: local arm selection:* For a sub-optimal arm $i$, we first upper bound the number of observations made by agents in $\mathcal{A}_i$, which is $n_t(i)$. Also, we reuse $\hat{n}_t^j(i)$ as the total number of observations received by agents $j$ up to $t$. Then, we have

$$n_t(i) \leq \hat{n}_t^j(i) + \sum_{j \in \mathcal{A}_i} \min \left\{ d_j \theta_j, n_t^j(i) \right\}, \ \forall \, j \in \mathcal{A}_i, \tag{15}$$

where on the right hand side, the second term refers to an upper bound of the number of outstanding observations, i.e., the observations that have not been received due to the delay between agents and $n_t^j(i)$ is number of observations maded by agent $j$. We also consider two types of decisions: Type-I corresponds to the decisions of an agent when the mean values of all arms lie in the confidence intervals calculated by the agent; and Type-II decisions refer to others. First, we focus on the cases that the algorithm makes a Type-I decision at time $t$, i.e., the mean value of any arm lies in its confidence interval calculated by agent $j$. Then, at time $t$, if agent $j$ in $\mathcal{A}_i$ selects arm $i$, we have

$$\sqrt{\frac{2\alpha \log \delta_t^{-1}}{\hat{n}_t^j(i)}} + \sqrt{\frac{2\alpha \log \delta_t^{-1}}{\hat{n}_t^j(i^*)}} \geq \Delta_i. \tag{16}$$

Otherwise, there is

$$\hat{\mu}(i, \hat{n}_t^j(i)) + \sqrt{\frac{\alpha \log \delta_t^{-1}}{2\hat{n}_t^j(i)}} \leq \mu_i + 2\sqrt{\frac{\alpha \log \delta_t^{-1}}{2\hat{n}_t^j(i)}} < \mu_i + \Delta_i - \sqrt{\frac{2\alpha \log \delta_t^{-1}}{\hat{n}_t^j(i^*)}}$$

$$= \mu_{i^*} - \sqrt{\frac{2\alpha \log \delta_t^{-1}}{\hat{n}_t^j(i^*)}} \leq \hat{\mu}(i^*, \hat{n}_t^j(i^*)) - \sqrt{\frac{\alpha \log \delta_t^{-1}}{2\hat{n}_t^j(i^*)}},$$

implying that arm $i$ is strictly dominated by $i^*$, hence, $i$ can not be selected by agent $j$, contradicting the assumption that $i$ is selected by $j$. It follows from Equation (16) that

$$\max \left\{ \sqrt{\frac{2\alpha \log \delta_t^{-1}}{\hat{n}_t^j(i)}}, \sqrt{\frac{2\alpha \log \delta_t^{-1}}{\hat{n}_t^j(i^*)}} \right\} \geq \frac{\Delta_i}{2}.$$

Thus, we have

$$\min \left\{ \hat{n}_t^j(i), \hat{n}_t^j(i^*) \right\} \leq \frac{8\alpha \log \delta_t^{-1}}{\Delta_i^2}. \tag{17}$$

Again, we define $Q$ as the number of Type-II decisions. Then, by Lemma 1 that is still valid for the delayed system, we have

$$\mathbb{E}\left[Q\right] \leq 2 \sum_{j \in \mathcal{A}} \sum_{l=1}^{N_j} \sum_{i \in \mathcal{K}_j} \frac{l\Theta_i}{\theta_j} \delta_{l/\theta_j}^{\alpha}. \tag{18}$$

By combining Equations (17) and (18), we have

$$\min\left\{\mathbb{E}\left[\hat{n}_T^j(i)\right], \mathbb{E}\left[\hat{n}_T^j(i^*)\right]\right\} \leq \frac{8\alpha \log \delta^{-1}}{\Delta_i^2} + \mathbb{E}\left[Q\right]. \tag{19}$$

Then, using Equation (15), we have

$$\begin{aligned}
\mathbb{E}\left[\hat{n}_T^j(i^*)\right] &\geq \mathbb{E}\left[n_T(i^*)\right] - \sum_{j \in \mathcal{A}_{i^*}} d_j \theta_j \geq \frac{T\Theta_{i^*} - \mathbb{E}\left[Q\right]}{K} - \sum_{j \in \mathcal{A}_{i^*}} d_j \theta_j \\
&\geq \frac{\Theta_{i^*} n_T(i)}{\Theta_i K} - \frac{\mathbb{E}\left[Q\right]}{K} - \sum_{j \in \mathcal{A}_{i^*}} d_j \theta_j,
\end{aligned} \tag{20}$$

where the second inequality is based on the fact that the expected number of decision rounds with the optimal arm in the candidate set is at least $T\Theta_{i^*} - \mathbb{E}\left[Q\right]$, and the third inequality is based on the fact that $T \geq n_T(i)/\Theta_i$.

Last, combining the results in Equations (15), (19), and (20), we get

$$\begin{aligned}
\mathbb{E}\left[n_T(i)\right] &\leq \max\left\{\frac{8\alpha \log \delta^{-1}}{\Delta_i^2} + F_i, \frac{8\alpha K\Theta_i \log \delta^{-1}}{\Theta_{i^*}\Delta_i^2} + K\frac{\Theta_i}{\Theta_{i^*}} \sum_{j \in \mathcal{A}_{i^*}} d_j \theta_j\right\} + \left(1 + \frac{\Theta_i}{\Theta_{i^*}}\right) \mathbb{E}\left[Q\right] \\
&\leq \max\left\{\frac{8\alpha \log \delta^{-1}}{\Delta_i^2} + F_i, \frac{8\alpha K\Theta_i \log \delta^{-1}}{\Theta_{i^*}\Delta_i^2} + KD\Theta_i\right\} + \left(1 + \frac{\Theta_i}{\Theta_{i^*}}\right) \mathbb{E}\left[Q\right],
\end{aligned}$$

where

$$F_i = \sum_{j \in \mathcal{A}_i} \min\left\{d_j \theta_j, \frac{8\alpha \log \delta^{-1}}{\Delta_i^2}\right\}.$$

*Case II: external arm selection:* Now, we aim at upper bounding the expected number of selection times for a suboptimal arm $i$ by the agents outside set $\mathcal{A}_i$. Again, we assume that agent $j$ makes a Type-I decision at time slot $t$. Consider the case that $I_t^j = i$ and $i$ is not within $\mathcal{K}_j$. By algorithm rules, we have that arm $i$ has the largest lower confidence bound. We prove that the following inequality must hold in this case.

$$\sqrt{\frac{\alpha \log \delta_t^{-1}}{\hat{n}_t^j(i^*)}} \geq \frac{1}{2}\Delta_i, \tag{21}$$

Otherwise, we have

$$\begin{aligned}
\hat{\mu}(i, \hat{n}_t^j(i)) - \sqrt{\frac{\alpha \log_t \delta_t^{-1}}{2\hat{n}_t^j(i)}} &\leq \mu(i) = \mu(i^*) - \Delta_i \\
&< \hat{\mu}(i^*, \hat{n}_t^j(i^*)) + \sqrt{\frac{\alpha \log \delta_t^{-1}}{2\hat{n}_t^j(i^*)}} - 2\sqrt{\frac{\alpha \log \delta_t^{-1}}{2\hat{n}_t^j(i^*)}} \\
&= \hat{\mu}(i^*, \hat{n}_t^j(i^*)) - \sqrt{\frac{\alpha \log \delta_t^{-1}}{2\hat{n}_t^j(i^*)}},
\end{aligned}$$

contradicting the rules of the algorithm.

Thus, at time $t$, the selected arm $i$ satisfies

$$\Delta_i \leq 2\sqrt{\frac{\alpha K \log \delta_t^{-1}}{(t-D)\Theta_{i^*} - Q}}.$$

The above equation is obtained by replacing $\hat{n}_t^j(i^*)$ in Equation (21) with $[(t-D)\Theta_{i^*} - Q]/K$, since $\hat{n}_t^j(i^*) \geq [(t-D)\Theta_{i^*} - Q]/K$.

For any agent $j$, the largest time slot when the agent makes a Type-I decision and a suboptimal arm $i$ lies in the candidate set is

$$4\frac{\alpha K \log \delta^{-1}}{\Delta_i^2 \Theta_{i^*}} + D + \frac{Q}{\Theta_{i^*}}.$$

Then, the regret spent on the arm $i$ in other agents is upper bounded by

$$4\frac{\alpha K \log \delta^{-1}}{\Delta_i} \frac{\Theta_i}{\Theta_{i^*}} + D\Theta_i + \frac{Q\Theta_i}{\Theta_{i^*}} + Q.$$

Summing up the above two pieces of regret and the expected number of Type-II decisions yields

$$\mathbb{E}\left[R_T\right] \leq \sum_{i:\Delta_i>0} \max\left\{\frac{4\alpha \log \delta^{-1}}{\Delta_i}\left(2 + \frac{K\Theta_i}{\Theta_{i^*}}\right) + F_i\Delta_i, \frac{12\Theta_i\alpha K \log \delta^{-1}}{\Delta_i\Theta_{i^*}} + KD\Theta_i\Delta_i\right\}$$

$$+ 2\left(1 + \frac{\Theta_i}{\Theta_{i^*}}\right)\sum_{j\in\mathcal{A}}\sum_{l=1}^{N_j}\sum_{i\in\mathcal{K}_j} \frac{l\Theta_i}{\theta_j}\delta_{l/\theta_j}^{\alpha} + D\Theta.$$

This completes the proof.