# OpenReview forum: "Cooperative Stochastic Bandits with Asynchronous Agents and Constrained Feedback"
_NeurIPS.cc/2021/Conference — NeurIPS 2021 Poster_

### Official Review · Reviewer_BHsX · 2021-07-13

**Rating:** 7
**Confidence:** 4

**Summary:**

This paper studies the problem of asynchronous multi-agent bandits when the agents have different local set of arms. The agents can pull any arm but only observe the feedback when pulling among their local set.
The authors propose both a lower bound and an algorithm with an upper bound scaling as the lower with an extra K multiplicative term.

**Limitations And Societal Impact:**

Not concerned

**Main Review:**

Post-rebuttal: I want to thank the authors again for their detailed answers. I now clearly understand why using LCB is better than 'best empirical arm' during the exploitation phase.

Concerning the lower bound, this would require a substantial modification of Theorem 1 (and its proof) and I would be grateful to the authors for communating the revised exact formulation of Theorem 1 as soon as available.
Overall, I do trust the authors for carefully correcting Theorem 1, thus raising my score.

------------------------------------------

The paper is well written and very clear. The ideas involved by this new setting are interesting and I thus think this paper is the ground for a great piece of work. Still, I have some small reservations concerning the proposed lower bounds and upper bounds, as I believe that both (and especially the latter) can be improved. I detail how below and let the authors answer in the rebuttal whether such improvements are indeed possible.

About the lower bound: first I think the Theta/Theta_i* should be put in the left term instead, as it goes to infty. Otherwise the Omega(.) notation is a bit weird.
More importantly, I would have hoped such a bound to hold even when Theta/Theta_i* remains fixed. I think it is a major drawback in the current lower bound and the difficulty to prove a similar bound for a fixed ratio should be discussed.

About the upper bound. First the presentation as an AAE algorithm with stages 1 and 2 is a bit awkward, since it seems that the set of active arms C_{j,t} is not necessarily decreasing in t. In particular, an inactive arm at some time step can be reactivated, as delta_t decreases, and thus the confidence interval grows. As a consequence, the current description of the algorithm is a bit counterintuitive and might be adjusted in a future version.

I also think the algorithm can be improved (and eventually the extra K factor removed) with minor modifications. To do so, it seems that if agents pull the maximal empirical arm (instead of maximal LCB) in stage 2, then the regret caused by the second stage should be constant in T. (to prove such things, you can for example use a proof with ideas similar to 'Optimal Algorithms for Multiplayer Multi-Armed Bandits' by Wang et al.) If this is not a good idea, it should at least be discussed in the paper as to me the instinctive way to 'exploit' during stage 2 is to pull the best empirical arm, not the largest UCB.

in stage 1, it seems that the K factor appears because the 'exploring' agent only pulls the optimal arm at least 1/K of her time. If instead she pulled her best (local) empirical arm with proba 1/2 and explored with proba 1/2, this might do the trick (at least for a single unique arm). (you can also adapt ideas from Wang et al. here I think)

------- minor comments -------

- Although being delayed to the Appendix, the related work section seems rather complete in my opinion. I just think the recent paper by Della Vecchia and Cesari is missing as a reference for asynchronous multi-agent bandits.

- the activation rounds of each agent are here deterministic. Why not use stochastic activation instead? (as done in sleeping bandits and in the existing literature on asynchronous multi-agent bandits)

- line 119: it seems you never exactly say that x_t(i) follows a (subgaussian) distribution of mean mu(i)

- Algo 1: lines 3-5 -> it is a bit weird to repeat these lines twice at each round. I think it should be better handled for presentation purpose


**Time Spent Reviewing:**

6 hours

---

> ### Author Response · Authors · 2021-08-10
> **Several technical responses regarding lower and upper bounds, and presentation issues of the algorithms**
>
> ## (on the lower bound)
> First, we agree that the factor $\Theta/\Theta_{i^*}$ should be placed on the left-hand side of the equation in Line 217. Second, we agree on the point that the bound still holds for the large constant of  $\Theta/\Theta_{i^*}$. Specifically, the lower bound is proved by using the fact in Line 601 which holds for large $\Theta/\Theta_{i^*}$. In other words, from the equation in Line 601, we can derive the range of $\Theta/\Theta_{i^*}$ with which the lower bound holds. Though, it is a little complicated to derive the closed-form for the range. We will add a remark below Theorem 1 (the lower bound theorem) and discuss the complications of deriving the lower bound.
>
> ## (on the presentation of the algorithm)
> We thank the reviewer for pointing out the fact that the size of the candidate is not necessarily decreasing. This is indeed correct. We will clarify this in the paper by clearly mentioning that the candidate set is constructed dynamically, and it is possible that after pulling some external arms, some local arms are put back into the local candidate set. That said, we believe that the current high-level presentation logic of the algorithm makes sense, since it shows the high-level idea of our algorithmic approach on focusing sufficiently on the local arm, and then moving to the stage of selecting external arms. This transition is not immediate and may take time due to the point that is highlighted by the reviewer.
>
> ## (on the K factor in the upper bound)
> In the opinion of the authors, the $K$ factor cannot straightforwardly be dropped from the regret result.
>
> - First, we show the reason why we cannot drop the $K$ factor by some simple extensions of well-known bandit algorithms or the one suggested by the reviewer. The difficulty of removing $K$ from the regret is the *heterogeneity of aggregate action rate* for individual arms, which is defined as $\Theta_i$ in our paper.  Note that the work of *Wang et al* considers a homogeneous setting with *an equal action rate* for all arms. In our model, a small action rate may result in low confidence for the estimates of some arms, and it leads to large regret if we select an arm with a high estimate but low confidence.
>
> - Second, we provide an example as evidence for our conjecture that the $K$ factor is unavoidable in the regret upper bound. Recall that $\Theta_{i^*}$ is the sum of action rates of the agents containing the optimal arm. In some cases, $\Theta_{i^*}$ is much smaller than the empirical observation rate for $i^*$. Consider the case where there are K/2 optimal arms residing in the same agents. In this case, the agents cannot focus all observations on a single optimal arm, and can only allocate $2\Theta_{i^*}/K$ observations to a particular optimal arm on average. That is why we can only guarantee that the 'exploring' agent only pulls an optimal arm 1/K of her time in the worst case, which is also mentioned by the reviewer. In the extreme case when K/2 is very large, few observations can be allocated to an optimal arm, with the regret being arbitrarily bad. This implies that it is difficult to remove the K factor from the regret results shown in Theorem 2 or Corollary 1 (we lost $\tilde{\Delta}_i>0$ under the sum operator in Corollary 1, this is a typo and we will fix it). Based on the above observation, the authors conjecture that there are no algorithms achieving a regret independent of the number of arms. To prove our result is tight, one possible way is to improve the lower bound.
>
> - Despite the above facts, the authors believe there must be a better way to present the upper bound and lower bound. For example, it is useful to clearly show that $\Theta_{i^*}$ serves as a theoretical upper bound for the aggregate observation rate of the optimal arm, and its empirical value could be very small in some cases. Thus, we can define better parameters $\Theta_{i^*}$ to characterize the regret upper bound. For example, $\Theta_{i^*}$ can be redefined as the sum of action rates of the agents containing the optimal arm divided by the number of local arms. In this way, the $K$ factor will be absorbed into the new definition of $\Theta_{i^*}$ to some degree, and hopefully, the regret under the new definition will capture the features of the model better.
> ## (minor comments)
> - We sincerely apologize for not reviewing the paper by Della Vecchia and Cesari, despite our extensive literature review, we missed this work during the preparation of our paper. In a future revised version, we will cite this paper and add it to our literature review.
>
> - The stochastic action (activation) rate is an interesting follow-up direction and we highlight it as future work.
>
> - We will carefully fix typos including both mentioned by the reviewer and proofread the paper.

---

> > ### Comment · Reviewer_BHsX · 2021-08-17
> > **Thanks for your answer**
> >
> > I first want to thank the authors for thoroughly answering to all the different reviews. Yet I still believe that the K term in the upper bound of the regret could be removed. Unfortunately, the authors did not answer (in my opinion) why the two suggestions I originally proposed should not work.
> >
> > - Although the rate $\Theta_{i^*}$ can be very small, if the exploiting players pull the best empirical arm, the number of exploiting pulls of arm $i$ should scale with $\frac{\Theta_i}{\Theta_{i^*}\Delta_i}$, thus being indepent from $T$. Wang et al. indeed use a synchronous model, but I still believe that their techniques can be easily adapted to this work
> >
> > - Concerning the second point in the paragraph (on the K factor in the upper bound), I agree that the case of multiple (eg $K/2$) optimal arms would require more carefulness and might be more intricate, which is why I precised "at least for a single unique arm" in my original review.
> >
> > If the authors convince me that the proposed ideas are actually bad, I will reconsider my score. In particular, even if the former leads to a similar regret bound, I think it might be used in the algorithm as pulling according to "best empirical" is simpler, more intuitive and empirically better than pulling according to LCB.
> >
> > ------------
> >
> > After the rebuttal, I also had another look at the proof of Theorem 1 and some points in its proof (in the second part) are now very confusing to me.
> >
> > - First the authors assume that Equation (13) holds as soon as the regret does not scale linearly with $\frac{\Theta}{\Theta_{i^*}}$. I do not see why this is true, as the regret could for example scale in $\frac{\Theta}{\Theta_{i^*}} \times \log\left(\frac{\Theta}{\Theta_{i^*}}\right)^{-1}$.
> >
> > - Even if this held, the second part of the proof of Theorem 1 only proves that the regret should scale linearly with $\frac{\Theta}{\Theta_{i^*}}$, when all other parameters ($\Delta, K, \ldots$) are fixed. But I do not see why the exact scaling in all parameters should then be as given by the equation line 605.
> >
> > I am sorry to only note these points now and hope that the authors will still be able to clarify the proof of Theorem 1.

---

> > > ### Author Response · Authors · 2021-08-20
> > > **Thanks for follow-up comments.**
> > >
> > > We first appreciate the time and effort of Reviewer BHsX for detailed comments and additional ideas for potential improvement of our paper. In the following, we provide our detailed response to each comment.
> > >
> > > > **Comment:** Although the rate $\Theta_{i^*}$ can be very small, if the exploiting players pull the best empirical arm, the number of exploiting pulls of arm should scale with $\frac{\Theta_i}{\Theta_{i^*}\Delta_i}$, thus being indepent from $T$. Wang et al. indeed use a synchronous model, but I still believe that their techniques can be easily adapted to this work.
> > >
> > >
> > >
> > > We show that selecting the external arm based on the "maximal empirical arm" policy in the second stage (as suggested by the reviewer) leads to a linear regret in $T$. Instead, the maximal LCB as index policy leads to a better regret result.
> > > To show this, consider an extreme case where there exists a very slow agent $A$ that samples only at the first time slot. Also, agent $A$ contains a single local suboptimal arm $\alpha$, whose reward subjects to a Bernoulli distribution with reward mean $0<p<1$, smaller than that of the global optimal arm. Since at most, only one observation of arm $\alpha$ is available, with probability $p$, the observed reward of $\alpha$ is $1$. Information on $\alpha$ will be broadcast by agent $A$ to other agents. Now consider there are multiple other agents that are in the second stage (due to some other good external arms that dominate their local arms). Following the "best empirical reward" policy, these agents, with probability $p$, select arm $\alpha$ with just one single observation, receive a suboptimal reward. Since the reward is not observable the empirical reward does not change, hence, this process iterates in their future sampling, and a linear regret in stage 2 will be incurred. We note that our proposed selection policy, i.e., the largest LCB, does not have such a drawback, since, by a single observation, the lower confidence bound of arm $\alpha$ is not shrunk enough, and hence, the agents in the second stage, will pull some other external (and well-observed) external arms with the largest LCB.
> > >
> > >
> > >
> > > > **Original comment:** in stage 1, it seems that the K factor appears because the 'exploring' agent only pulls the optimal arm at least 1/K of her time. If instead she pulled her best (local) empirical arm with proba 1/2 and explored with proba 1/2, this might do the trick (at least for a single unique arm). (you can also adapt ideas from Wang et al. here I think)
> > >
> > > > **The follow-up comment in the discussion:** Concerning the second point in the paragraph (on the $K$ factor in the upper bound), I agree that the case of multiple (eg $K/2$) optimal arms would require more carefulness and might be more intricate, which is why I precised "at least for a single unique arm" in my original review.
> > >
> > >
> > > For the "exploring" agents in stage 1, we admit there is still room to improve their empirical performance by guaranteeing more observations on the optimal arm, especially when the number of global optimal arms is much less than $K$ (which is a less restrictive assumption than having a unique optimal arm).
> > > Our policy can only guarantee at least $\Theta_{i^*}/K$ observations of the optimal arm and might be slow in eliminating suboptimal arms, and consequently, a $K$ factor is introduced to the regret bound.
> > > In the special case of having a unique optimal arm, the policy suggested by the reviewer (close to Wang’s policy), i.e., assigning half observations on the best (local) empirical arm, sounds like a promising alternative.
> > > Though, when the unique optimal arm is in the local subset of multiple agents with different action rates, the problem becomes more complicated and needs more effort to implement this idea.
> > > To conclude, the suggestion for changing the policy in stage 1 makes much sense and sounds very interesting to the authors as an initial idea to drop the $K$ factor. However, it comes with technical challenges that are unique to our problem setting. We believe new results can be obtained using this idea as follow-up work.
> > >
> > >
> > > In the following, we proceed to respond to the second category of comments. We appreciate the new comments on the proof of the lower bound results. We admit that both comments on the lower bound are valid. Our plan is to revise the statement of Theorem 1 to address the issue that is raised in the first comment. The second comment could be addressed by some additional modifications in the proof steps, however, we believe that with these modifications we can still obtain the same lower bound result. We now provide detailed responses and our plan for the next revised version.
> > >
> > >
> > > > **Comment:** First the authors assume that Equation (13) holds as soon as the regret does not scale linearly with $\frac{\Theta}{\Theta_{i^*}}$. I do not see why this is true, as the regret could for example scale in $\frac{\Theta}{\Theta_{i^*}} \times \log \left(\frac{\Theta}{\Theta_{i^*}} \right)^{-1} $.
> > >
> > > It is true we can not prove the lower bound which scales linearly with $\frac{\Theta}{\Theta_{i^*}}$ when Equation (13) does not hold.
> > > This issue can be solved by restating the results on the lower bound in Theorem 1. Specifically, we plan to rewrite the statement of Theorem 1 to show that there is no $\sigma<1$ with which the regret of any algorithm is $O((\Theta/\Theta_{i^*})^{\sigma'})$.
> > >
> > >
> > > > **Comment:** Even if this held, the second part of the proof of Theorem 1 only proves that the regret should scale linearly with $\frac{\Theta}{\Theta_{i^*}}$, when all other parameters ($\Delta$, $K$, ...) are fixed. But I do not see why the exact scaling in all parameters should then be as given by the equation line 605.
> > >
> > > In the second-part proof of Theorem 1, and for the sake of easier understanding of the logic, we dropped the system parameters, such as $K$, $\Delta$ when proving the dependency on $\Theta/\Theta_{i^*}$. To make the reasoning more clear, we plan to make the following modifications.
> > >
> > > First, we plan to rewrite (13) as follows.
> > >
> > > $$\limsup\limits_{T\rightarrow +\infty,\frac{\Theta}{\Theta_{i^*}}\rightarrow +\infty}\frac{R_{T}(\pi,\nu)}{(\Theta/\Theta_{i^*})^{\sigma'}\log(T\Theta)}=O\left(\sum_i\frac{\Delta_i}{\text{KL}(P_i,P'_i)}\right). $$
> > > The new version of (13) contains the above-mentioned parameters and by similar reasoning, we can prove that the above equation does not hold for any $0<\sigma'<1$ and complete the proof. In the next revised version of the paper, we will carefully proofread the details of the proofs and make sure that the entire logic and details of the proof are concrete.

---

### Official Review · Reviewer_b73Y · 2021-07-15

**Rating:** 7
**Confidence:** 4

**Summary:**

This paper considers a variant of the cooperative bandits where there are N agents cooperatively trying to find the best arm, but have certain constraints. First, each agent has a "rate" that is the number of time-steps after pulling an arm after which they can pull another arm. This rate can be different across different agents. Second, each agent has a subset of arms that are local which it can pull and see the observation and the rest are global which it can pull and only receive the feedback via communication from other agent. The broadcast information is assumed to be 0 cost and 0 delay. The goal of the agents is to minimize the total regret across the T time-steps. For this new setting of the problem, the paper provides an active arms elimination style algorithm that keeps playing the local arms, while collecting the broadcast for other global arms, until the confindence interval of some local arm does not overlap with that of a global/local arm and that the mean is lower.

**Limitations And Societal Impact:**

Overall I do not find much limitations to the work in this paper. One suggestion would be to take a concrete application and described in detail how this setting can model that problem. I do not find this to be a major point, but just a minor comment on the presentation so that the results are widely appreciated in the conference.

Regarding societal impact, since this is primarily a theory paper I do not forsee much impact beyond the general impact bandits as a field has.

**Main Review:**

Originality: The problem introduced in this paper is new and introduces a new twist to the cooperative bandit setting. This twist is well-motivated by practical applications where each agent can be thought of as a server in a distributed setting and the only communication with requests served by other servers is via the internal network. The algorithm considered in this paper is also novel and does not do the usual "construct UCB and pick the largest arm" type approach. In fact, the idea of playing only locally available arms and getting the feedback for the global arms via the network seems to be original. The paper also confirms their theory with simulations.

Quality/Clarity: Overall the paper is well-written and clear to read. The paper explains the setup clearly, their algorithm is also described in detailed informally and overall very pleasant to read and understand. The simulations in the paper also are useful add and complements the theory thus, giving more confidence on the correctness of the main theorem without having to carefully check all the steps in the proof.

Significance: Since the problem is new the results obtained in this paper are new and significant. Moreover, the notion of public and private arms is an interesting dimension and is of potential interest beyond cooperative bandits. For instance, an interesting future direction is to study the same in the competitive environment. Here the dynamics of the agents will be completely different, and an agent will be incentivized to play global arms, even if they don't see feedback to "block" other agents.


**Time Spent Reviewing:**

5

---

> ### Author Response · Authors · 2021-08-10
> **Response to Reviewer b73Y**
>
> We greatly appreciate the positive comments from the reviewer and will work our best to improve the paper in the revised version to make it better. As suggested by the reviewer, in the revised version, we will provide a concrete application that can be captured by our setting. We already mentioned *a few initial examples* (online shortest path routing in virtual private networks, clinical trial, and crowdsourcing) in our response to Reviewer rnQi, and will work on them more rigorously to pick the one that is more reasonable and concrete.

---

### Official Review · Reviewer_rnQi · 2021-07-23

**Rating:** 5
**Confidence:** 4

**Summary:**

The paper studies a cooperative multi-agent stochastic bandit problem with a setting in which each agent only have limited access to a local subset of arms and all the agents are asynchronous with different gaps between decision-making rounds. An algorithm is proposed for this setting and a regret bound is derived.

**Limitations And Societal Impact:**

Yes

**Main Review:**

The paper considers important aspects of cooperative multi-agent bandit problems, one is the heterogeneous setting in which each agent only has access to a subset of all arms, and the other is asynchronous sampling. Both two aspects have not been (fully) studied in the literature.

My main concern of the paper is its problem formulation. It is assumed that the agents are allowed to pull and receive a reward from any arm but only receive observations from local arms. That is to say, an agent may pull an arm without knowing the reward. Is there any application for such a setting? Meanwhile, it is also assumed that once each pulls an arm, it instantaneously forwards this observation to all other agents. Therefore, each agent can constantly be aware of all reward sampling information in the network, except for some of its own rewards. Since there is no reward distribution difference among the agents, the setting and assumptions imply that each agent can learn the best arm simply using the standard single-agent stochastic bandit algorithms. Although the asynchronous sampling brings difference and possibly complication, the over setting/assumption for the heterogeneous observability is not convincing and lacks applications.

Update Sep 2: Thanks the authors for very detailed responses. After intensive discussion with other reviewers, I better understand the main technical contribution of the paper: improve the regret dependency on action rates in the asynchronous case. With this, I increase my score from 4 to 5. I suggest the authors to move the delay case results to the main paper, as it can deal with general non-incomplete graphs. The current main paper focuses on complete graphs, which is quite limited for a cooperative setting. My remaining concern is when dealing with non-complete graphs, the paper allows reward relays over the network, which simplifies the algorithm design and analysis. Another comment is that the current lower bound relies on a limiting condition, which is not a general lower bound.

**Time Spent Reviewing:**

6

---

> ### Author Response · Authors · 2021-08-10
> **A few relevant applications and significance of the technical contributions**
>
> ## (on the motivating application scenario)
> There are two new features in our multi-agent bandit setting: asynchronous sampling and partial access to arms. The asynchronous sampling has clear practical motivation due to the nature of multi-agent (or distributed) decision-making. In the following, we focus on the justification for the latter on partial access to arms. We highlight three different examples in the context of the online shortest path routing (OSPR), clinical trials, and crowdsourcing, which are classical applications of bandits. *We note that we do not claim that our current model fully captures the following examples; instead, it provides initial modeling with interesting and nontrivial mathematical challenges that are addressed in this paper. This initial model, then, can be extended to fully capture real applications.*
>
> 1. **OSPR:** In the basic version of OSPR, a bandit algorithm could be applied to select a path (arm) with minimum delay. Now consider an extended version of OSPR in a heterogeneous network that includes multiple virtual private networks (VPN) each represented by an agent (or a gateway, in networking terminology).  In this scenario, the local arms of an agent represent the paths including the link in their VPN, and the paths in other VPNs are external arms. In the case of selecting a path that goes through some nodes in another VPN, the path information might not be observable to the rest of the network, which resides outside of the VPN, i.e., other agents. This scenario could be captured by our bandit setting in which an external arm (a path in another VPN) is selected and the reward is allocated (the delay), but it is not observable.  On the other hand, when an agent selects a path in its local VPN, they can share the information with the representative agents of other VPNs to eventually find the globally shortest path in the entire network.
>
>
>
> 2. **Clinical Trial:** A doctor in a clinic is required to offer a treatment plan for patients with some disease, while the clinic, which the doctor sits in, can only provide partial options due to lack of medical equipment. The goal of the doctor is to maximize his reputation or the number of treatment successes. In the medical decision-making setting, the plan for the disease corresponds to the arm, and the doctor in the clinic is an agent. If the doctor decides on a specific treatment option that is not offered in their clinic, the patient may not come back for follow-up appointments, and in this case, the patient’s medical record or feedback might be missing. Thus, the treatment plans offered by the clinic are the local arms in our model. Cooperation among the doctors in different clinics helps to accelerate learning the best treatment plan, by sharing medical statistics.
> 3. **Crowdsourcing:** In crowdsourcing, workers are allowed to register to any agents, each of which is tailored for specific tasks. An agent maintains the profile of registered workers, with a reputation score (e.g., the success rate of finishing a job) updated with their performance in satisfying tasks. In real-world applications, workers are allowed to be anonymous to some agents with the aim to protect personal privacy, with only the reputation score revealed to the agent. For anonymous workers, the agent treats them as guests and no identity or profile can be tracked for them. Hence, the observed reward is useless. In crowdsourcing with multiple agents, anonymous workers correspond to external arms, and registered workers correspond to local arms. In this model, an agent selects anonymous workers only based on reputation scores by other agents. Implicitly, different agents cooperate by maintaining the common reputation score for registered workers.
>
> It is worth noting that, to the best of our knowledge, our work is the first that tackles a multi-agent bandit setting where agents have access to a subset of arms. The basic setting that we considered in this paper, however, could be extended to better capture more convincing practical applications. We highlight a few practically relevant, and (to our belief) feasible extensions. First, our model could be extended to the case that instead of exact information, a perturbed, yet useful observation is communicated between agents. This extension makes the setting much more interesting from a practical perspective since in this setting agents can share some limited information with others to incentivize them to use their local resources (arms). Another practical extension is adding the communication delay in the model as we tackled it in Appendix D of our supplementary material. One can consider communication costs in the model and the goal becomes to provide low regret algorithms with low communication costs. As the last note regarding another potential extension of our model, we refer to the “significance” comment of Reviewer b73Y:
>
> > “Moreover, the notion of public and private arms is an interesting dimension and is of potential interest beyond cooperative bandits. For instance, an interesting future direction is to study the same in the competitive environment. Here the dynamics of the agents will be completely different, and an agent will be incentivized to play global arms, even if they don't see feedback to "block" other agents.”
> Hence, we see great potential for follow-up and more practical works on the topic of multi-agent bandits with limited access to arms.
> ## (on the possibility of using standard single-agent stochastic bandit algorithms)
> In the current manuscript, there are multiple sections that we explained why the standard single-agent stochastic bandit algorithms fail to achieve a good performance in the setting of our work. In the following, we refer to specific sections of the submitted paper and also the comments by other reviewers.
> 1. A high-level justification for the failure of standard algorithms is given in Line 53-63 (the third paragraph on the 2nd page). In short, we highlight that UCB-like algorithms fail to effectively deal with the low sampling rate of suboptimal arms located in slow agents. Hence, asynchronous sampling is a challenge that emphasizes new algorithm design.
> 2. More technical reasoning on the failure of standard algorithms is given in Sections 3 and 4. First, in Section 3.3 (Lines 191-208), we present how the standard algorithms (UCB and AAE) could be extended to be applicable to our setting. Then in Section 4 (Lines 227-234), the failure of standard algorithms is investigated in terms of dependence on any slow agent, and more technical analysis is given in Section C.4 of the supplementary material (Lines 606-624). In short, our theories show that the regret of a simple extension of AAE depends on the sampling rate of any suboptimal arm located in slow agents. Instead, the regret of our proposed algorithm depends on the sampling rate of the agents including the optimal arm.
> 3. The intuitions and technical analysis on the failure of standard approaches are further verified by numerical experiments, as shown in Figure 1. The results clearly show that our proposed algorithm outperforms UCB and another alternative.
> 4. The reviewer can also refer to the “originality” comment by Reviewer b73Y that says:
> > “ The algorithm considered in this paper is also novel and does not do the usual "construct UCB and pick the largest arm" type approach. In fact, the idea of playing only locally available arms and getting the feedback for the global arms via the network seems to be original.“
>
> Putting together the above arguments, the current setting, even with the assumption of *“no reward distribution difference among the agents”*, adds nontrivial and mathematically interesting challenges for designing new bandit algorithms.

---

> > ### Comment · Reviewer_rnQi · 2021-08-28
> > **Re: A few relevant applications and significance of the technical contributions**
> >
> > Thanks the authors very much for detailed response.
> >
> > As the authors admitted, the proposed model is related to those possible application scenarios, yet the direct applications of the model to those scenarios are not obvious. This is fine. I agree that there is always a gap between theoretical models and real applications.
> >
> > The partial observation of the arm set is a natural and interesting problem for multi-agent bandits. After carefully reading the paper and existing cooperative bandit papers again, my concern of the model is further detailed as follows.
> >
> > Consider the existing synchronous distributed bandit settings in which all the agents have the full observation of the same set of arms. Each agent can receive information (usually not the reward itself for immediate, lower-level privacy concerns) with certain other agents. The information flow structure is described by a possibly sparse connected graph. Existing algorithms can solve this setting. Now suppose each agent can only observe the rewards of a subset of the original arm set, but the union of their observable subsets equals the original set. In this case, each agent can still send one arm, say arm k, information to its neighbor, say agent j, even though arm k is unobservable to agent j. It is not hard to see that the existing algorithms can directly work for such a partial observability scenario, mainly because there is no restriction in reward information transmission. Note that it still works for sparse connected graphs. Of course one can expect the learning procedure will be slower compared with the full observability scenario, since each agent may not have its own reward information from time to time. One more step, assume each agent may not activate every time step and can decide its own independent active time sequences, which leads to a quite general discrete-time asynchronous setting. Since many existing synchronous distributed bandit algorithms are consensus-based, and discrete-time consensus is robust against asynchronous updating, I expect it won't be hard to prove the above partial observability algorithms can be extended to the asynchronous case. Such extensions have been successfully achieved in various distributed learning and optimization problems. Also note that this is for any connected graphs and quite general asynchronous updating. Compared with these, the paper under consideration deals with a complete graph (as each agent can immediately transmit its reward to all other agents) and a special asynchronous updating scheme (as each agent has its fixed action rate). In summary, the proposed partial observability scenario is quite interesting, but the proposed model for cooperative learning is a bit too simplified/special.
> >
> > Compared with the above model and possible (straightforward) solution, what are the possible advantages/challenges of the paper under consideration?
> >
> > After going through other reviews and responses, a new issue is there seems a "gap" between theoretical results and the experiments. Since the theoretical results claim that the proposed AAE-LCB depends only on the action rate of the optimal arm, it outperforms the baseline AAE-AAE which depends on the slowest arm. Thus AAE-LCB could have significant regret improvement when the slowest arm is set to be very slow. But the experiments only show that the improvement is slight. Is there any possible explanation for this?

---

> > > ### Author Response · Authors · 2021-08-30
> > > **Response to additional comments**
> > >
> > > We appreciate the time and effort of the reviewer for additional comments and feedback. In what follows, provide our response to the following question:
> > >
> > >
> > > > "Compared with the above model and possible (straightforward) solution, what are the possible advantages/challenges of the paper under consideration?"
> > >
> > > Our response is organized into three parts: we first highlight the unique challenge of our model that emphasizes the need for a new algorithm design. Then, we further clarify the details of our model that we believe are technically crucial and are not straightforward to be captured by the ideas proposed by the reviewer. Last, we highlight the advantages of our model.
> > >
> > > ## (Unique technical challenges of our model)
> > >
> > > The existence of local/external arms (partial observability) in our model naturally emphasizes a two-stage learning approach that is justified in the introduction of the submitted paper (see Lines 53-85) and the LCB policy for exploiting external arms (see lines 174-185). To the best of our knowledge, none of the prior algorithms for distributed bandits with an underlying topology graph can directly address these challenges. Our theoretical and experimental results clearly show the importance of designing a proper "first local then external" strategy for our model, and we are not aware of such policies in the existing literature for multi-agent bandits. As we have replied in our prior response to the comment "on the possibility of using standard single-agent stochastic bandit algorithms", these challenges are unique to our problem setting. Extending existing algorithms for multi-agent bandits without careful attention to the *local vs. external arm selection dilemma* will not lead to interesting theoretical results in our heterogeneous setting since we demonstrated that poor regret will be incurred by naively extending UCB-like policies (see Lines 53-85). In addition, in our response to reviewer BHsX, we showed the inefficiency of the "best empirical" policy for external arm selection in our model. This policy, however, is proved to be efficient in a synchronous setting (see "Optimal Algorithms for Multiplayer Multi-Armed Bandits"). Hence, the transition from a synchronous and asynchronous model could be nontrivial.  More details could be found in our response to Reviewer BHsX.
> > >
> > > ## (Further clarification of the settings of our model)
> > >
> > > We appreciate the suggested high-level ideas of the reviewer on how prior distributed or consensus-based algorithms could be potentially applied to our setting. However, we believe that the eventual execution and regret analysis of these ideas need to address the unique challenge of a proper policy for selecting a local or external arm. The current suggested ideas, however, are not sufficiently clear to us to scrutinize their potential for capturing this challenge.
> > > We also would like to point out a potential incomplete understanding or misunderstanding by the reviewer that may change their suggested ideas. In particular, the reviewer says:
> > >
> > > > "In this case, each agent can still send one arm, say arm $k$, information to its neighbor, say agent $j$, even though arm $k$ is unobservable to agent $j$."
> > >
> > > The above sentence, at best, partially captures the limited feedback (or partial observability, in the reviewer's terminology) scenario of our proposed model. The closest rephrase that is compatible with our model is: "In this case, each agent $j$ can pull an external arm, say arm $k$, from its neighbor, say agent $j'$, even though the reward of arm $k$ is unobservable to agent $j$." Due to this particular feature, each agent should make a decision between selecting a local arm with observable reward or selecting an external arm that might be better than local arms but without observing the reward.
> > > This rephrase might be the key to help the reviewer to understand the need for the two-stage learning approach proposed in this paper. Given this additional clarification, we would appreciate the reviewer if they can provide additional details on their ideas or name a few references such that we can make a more rigorous and concrete response on the feasibility of such ideas.
> > >
> > > ## (On the advantages of our model)
> > >
> > > The first question raised by the reviewer partially asks about the possible advantages of our model. Our work captures partial access to the global set of arms. To the best of our knowledge, the partial access to arms is not captured by the prior literature on multi-agent MAB research. This new consideration is practically relevant since it captures large-scale learning scenarios and several other applications as highlighted in our previous response to this reviewer. Moreover, this partial access is mathematically an interesting addition that leads to additional algorithmic challenges, e.g., on local vs. external dilemma; and algorithmic ideas such as two-stage learning and using LCB for selecting external arms.
> > >
> > > ## (on the gap between the theory and experiments)
> > > In what follows, we respond to the following comment by the reviewer.
> > >
> > > > "After going through other reviews and responses, a new issue is there seems a "gap" between theoretical results and the experiments. Since the theoretical results claim that the proposed AAE-LCB depends only on the action rate of the optimal arm, it outperforms the baseline AAE-AAE which depends on the slowest arm. Thus AAE-LCB could have significant regret improvement when the slowest arm is set to be very slow. But the experiments only show that the improvement is slight. Is there any possible explanation for this?"
> > >
> > > Yes, the current experimental results do not clearly demonstrate the potential significant gap between AAE-AAE and AAE-LCB. The reason is that the current experimental scenarios are not designed to show this theoretical worst-case performance gap. Instead, our major goal was to show how the proposed algorithms empirically perform using real traces as compared to two baselines: First, the simple extension of UCB-like algorithms (that is called CO-UCB in results) that do not take into account the two-stage learning paradigm. Second, the comparison between AAE-AAE and AAE-LCB, both proposed in this work, on the importance of a good policy for pulling external arms. As the most important observation, our experimental result clearly shows the significant improvement of our proposed two-stage algorithms (recall that AAE-AAE is also two-stage) against CO-UCB that do not distinguish between local and external arms.
> > >
> > > The second expected observation about the advantage of AAE-LCB against AAE-AAE is not clear due to the way that we constructed the experimental scenarios. While the theoretical results characterize the regrets in the worst case, the goal in our numerical results was to demonstrate those in the average case using real data traces. Toward this, in our experiments, we set two groups of agents: one fast and the other one is slow. From the main theorem, our algorithm outperforms the baseline algorithms only when the optimal arm lies in a fast agent. That is because the regret of the alternative algorithm (AAE-AAE) degrades with the action rate of the slowest agent, while that of AAE-LCB depends on the action rate of the agent which the optimal arm lies in. In our experimental scenarios, however, the optimal arm is randomly assigned to either slow or fast agents, and thus the numerical results are NOT capturing the worst-case theoretical instance that is needed to showcase the better regret of AAE-LCB as compared to that of AAE-AAE. That said, in the next revised version, we can straightforwardly add a synthetic scenario and generate a special worst-case instance to show this theoretical gap.

---

> > > > ### Comment · Reviewer_rnQi · 2021-09-02
> > > > **Re: Response to additional comments**
> > > >
> > > > Thanks the reviewers for detailed responses!
> > > >
> > > > I want to ask a clarification question after reading the following response:
> > > >
> > > > ["The above sentence, at best, partially captures the limited feedback (or partial observability, in the reviewer's terminology) scenario of our proposed model. The closest rephrase that is compatible with our model is: "In this case, each agent $j$ can pull an external arm, say arm $k$, from its neighbor, say agent $j'$, even though the reward of arm $k$ is unobservable to agent $j$." Due to this particular feature, each agent should make a decision between selecting a local arm with observable reward or selecting an external arm that might be better than local arms but without observing the reward.]
> > > >
> > > > Even though agent $j$ cannot observe the reward of arm $k$ by itself, it can still receive the reward information of arm $k$ from other agents. There are two possibilities. First, whenever any other agent pulls arm $k$, the reward will be immediately forwarded to agent $j$ (as stated in line 41). Second, any other agent can also send reward information (e.g., the sample mean so far) to agent $j$. In the clinical trial application mentioned in the authors' earlier response, the second case can be viewed as doctors/clinics exchange their information from time to time. Note that here the communication graph is assumed as complete, i.e., each doctor/clinic can send information to every other doctor/clinic. Thus, each agent can have all arms' reward information. Did I miss anything here?

---

> > > > > ### Author Response · Authors · 2021-09-02
> > > > > **Additional clarification and about moving the case with delay into the main body**
> > > > >
> > > > > Thanks again, no, nothing is missed here. The only clarification is that the second case you have mentioned will not add any additional useful information for agents in our model. Essentially, by the immediate and honest broadcast of the observed information from each agent to others, and without taking into account the delay, everyone will be on the same page in terms of empirical reward information of the arms. Hence, no need to further communicate as mentioned in the second possible option of the reviewer.
> > > > >
> > > > > The challenge for each agent then becomes whether to pull a local arm and add a valuable empirical observation that is useful to the entire group of agents, or to simply rely on the available observation of the others without adding any additional useful information for decision making in future. This is really the core challenge that is new to our model and we discussed it extensively in the prior discussions and also in the paper. We consider addressing this challenge as the major technical contribution of the paper.
> > > > >
> > > > > About moving the case with a delay into the main body; we honestly admit that our original plan was to include this into the main body of the paper. However, after several rounds of discussions, we decided to remove it from the main flow of the paper. The main reason is that the other two new modeling aspects, i.e., partial access and asynchronous sampling, were by far more important and challenging to deal with. In this way, the core contribution of the paper becomes more clear and easier to follow. We believe that our partial access and asynchronous sampling could be extended into several other challenging and practically interesting problems for the research in distributed and multi-agent bandits.

---

> > > > > > ### Author Response · Authors · 2021-09-02
> > > > > > **One last correction**
> > > > > >
> > > > > > One last correction in the statement of the reviewer. The rigorous statement of the first option mentioned by the reviewer should be as follows. The changes are highlighted in **bold**.
> > > > > >
> > > > > > First, whenever any other agent pulls **a local** arm $k$, the reward will be immediately forwarded to agent $j$ (as stated in line 41). **Also, no agent can observe the reward when pulling an external arm.**
> > > > > >
> > > > > > Thanks.

---

> > > > > > > ### Comment · Reviewer_rnQi · 2021-09-02
> > > > > > > **Re: One last correction**
> > > > > > >
> > > > > > > Thanks!  Reviewer BHsX has helped clarify this point.  I misunderstood it earlier.

---

### Official Review · Reviewer_LVPx · 2021-07-30

**Rating:** 6
**Confidence:** 4

**Summary:**

This paper studies an asynchronous distributed multi-arm bandit problem. In this problem, each agent can pull arms at a different rate. In addition, each agent can only observe the reward from a subset of arms. The reward observed by an agent is immediately broadcasted to all other agents. The question is to analyze regret. The paper provides both upper and lower bound results.



**Limitations And Societal Impact:**

Please see above. I feel it is mostly about being patient and carefully polish the first 9 pages.

**Main Review:**



Contribution:
	Modeling: I believe it is important to have a clean model for distributed multi-arm bandit problem, in which each agent can pull arms at a different rate. I think many that have worked on distributed multi-armed problems secretly wondered this question (but I have not been following the literature for a while). The model, algorithmic, and lower bound results are quite clean. I think this "co-optimization" between model and upper/lower bound needs to be appreciated, e.g., the most clean model may have ugly upper/lower bound, and vise versa.

Both the algorithmic and the lower bound results are non-trivial. Most of the intuition explained in the papers is clear. The experiment results are also nice proof-of-concept.

In terms of writing, I see an effort was made here but it still feels like a difficult parse. A few examples:

(i) When the authors describe intuition of the paper, they refer to prior works but related work section appears outside first 9 pages. Putting related work outside first 9 pages is a quite non-standard practice. It appears to be a symptom that the authors lost patience to carefully pack substance into 9 pages (just a speculation). Neurips becomes quite cut-throat in recent years and these "details" matter.

(i) The number of symbols is also quite large, resulting in additional obstacles in reading. Some questions (a) what is alpha? It looks difficult to find its definition, (b) I cannot quite immediately tell a factor of $K$ gap as claimed by the authors from lower and upper bounds (theorem 1 & 2) (c) What happens when $\Delta_i = 0$, it looks in upper bound, we get 0 divided by 0, and in lower bound, we get 0 in denominators (in Corollary).

Also, a gap of $k$ feels a quite serious problem. It smells like a trivial bound so some effort to explain this gap could be useful.



**Time Spent Reviewing:**

2 hours

---

> ### Author Response · Authors · 2021-08-10
> **Some clarifications regarding literature review and some technical issues**
>
> Thanks a lot for your positive comments on this paper and in the following, we provide our responses to your comments.
>
> ## (literature review)
> We admit that it is better to move the literature review into the main body of the paper. NeurIPS allows an additional content page for the camera-ready version of accepted papers. If our submission is accepted, we will move the related work section into the main body of the paper.
>
> ## (on the $K$ factor in bounds)
> We refer to our response to Reviewer BHsX for additional discussions. We also add a remark after Corollary 1 to clarify how a gap of factor $K$ appears in lower and upper bounds. In short, by replacing $\Theta_i$ with $\Theta$ in Corollary 1, it becomes easier to see that the upper bound approaches the lower bound up to a $K$ factor. We will clarify this more rigorously in the revised version.
>
> ## (A typo in Corollary 1)
> There is a typo in the sum term in Corollary 1, the sum should be taken over the positive values of the $\Delta_i$, i.e., for those arms that are suboptimal. This is a typo and we will fix it in the revised version.

---

### Decision · Program_Chairs · 2021-09-27

**Decision:**

Accept (Poster)

**Comment:**

Reviewers are mostly positive about this submission. They agree that the paper studies a new and relevant problem, and the techniques are non-trivial and interesting. The reviews also provide many directions for the paper to improve, e.g. (1) correcting and clarifying some technical details (2) adjusting the presentation order for better reader experience (3) strengthening the lower bound.